# Global near real-time 500 m 10-day FPAR dataset from MODIS and VIIRS for operational agricultural monitoring and crop yield forecasting

Lorenzo Seguini[1,2,*], Anja Klisch[3,*], Michele Meroni[4], Anton Vrieling[1], Giacinto Manfron[5], Clement Atzberger[6], and Felix Rembold[5]

[1]Faculty of Geo-Information Science and Earth Observation (ITC), University of Twente, Enschede, The Netherlands
[2]Engeneering, Roma, Italy
[3]ThüringenForst, Forestry Research and Competence Centre, Jägerstraße 1, 99867 Gotha, Germany
[4]Seidor Consulting, Barcelona, Spain
[5]European Commission, Joint Research Centre (JRC), Ispra, Italy
[6]CYCLOPS MRV, New York, United States
[*]These authors equally contributed to this work.

**Correspondence:** Lorenzo Seguini (l.seguini@utwente.nl), Felix Rembold (felix.rembold@ec.europa.eu)

**Abstract.** Climate change and extreme weather events pose challenges to food security, emphasizing the need for reliable and timely monitoring of crop and rangeland conditions. For this purpose, long-term consistent Earth Observation datasets on vegetation conditions are typically used in early warning and crop yield forecast systems. However, the near-real-time (NRT) production of high quality datasets and the need to guarantee long-term records present various challenges. To address these, we present a NRT global dataset of Fraction of Photosynthetically Active Radiation (FPAR) at 500 m resolution, optimized for agricultural applications. Our dataset combines MODIS-FPAR (Collection 6.1) and VIIRS-FPAR (Collection 2) data, ensuring continuity from 2000 to well beyond 2030. We applied a robust filtering approach based on the Whittaker smoother to produce reliable FPAR estimates in NRT, accounting for sparse and irregular spaced observations due to cloud cover. The dataset is composed of two 10-day filtered timeseries: 1) MODIS-FPAR for 2000 to 2023, being the reference dataset, and 2) intercalibrated VIIRS-FPAR for 2018 onward. While several methods can effectively smooth and gap-fill FPAR data (i.e., using observations before and after the estimation date), our method is designed for optimal filtering in NRT (i.e., using only prior observations). Our approach yields six successive estimates of the same FPAR data point with increasing quality: an inital estimate immediately after the 10-day reference period, four subsequent estimates every 10 days using new observations, and a final consolidated estimate 90 days later. The implemented filtering ingests the available FPAR observations and their original quality assessment (QA) layers. To avoid unrealistic extrapolation when observations are sparse, we impose constraints, season and location specific, to FPAR estimates. We then intercalibrated the VIIRS-FPAR with the MODIS-FPAR filtered timeseries, using a mean difference correction approach, to ensure consistency between both series. This paper describes the filtering and intercalibration method used, the quality assessment of resulting timeseries, and details the obtained products and the corresponding QA layers. The NRT FPAR dataset is publicly available through the Joint Research Centre Data Catalogue, https://data.jrc.ec.europa.eu/dataset/1aac79d8-0d68-4f1c-a40f-b6e362264e50 (Seguini et al., 2025).

## 1 Introduction

Climate variability and frequent extreme weather events result in reduced agricultural productivity, thus contributing to food price volatility, food insecurity, malnutrition, and global hunger (FAO and IFAD, 2020; Programme, 2022). Early warning systems (EWS) and crop yield forecasting systems (CYFS) use meteorological and Earth Observation (EO) data (Fritz et al., 2019; Nakalembe et al., 2021) to provide information on ongoing or potential issues in crop and rangeland production, to assess market implications and food security concerns. EO technologies are crucial for monitoring crop and rangeland conditions, providing biophysical data on vegetation over large areas with high revisit frequency (Atzberger et al., 2015).

Despite the increasing availability of free data from high-resolution optical sensors (e.g., from Landsat and Sentinel-2 missions), low-resolution sensors (250-1000m) remain valuable for their frequent revisits (and thus larger availability of cloud-free observations) and longer timeseries. The latter is of utmost importance for anomaly computation and crop yield forecasting. Anomalies compare the current crop conditions to long-term climatological statistics, while data-driven crop yield forecasting uses multi-year EO-based timeseries as predictors against crop yield statistics (Basso and Liu, 2019; Schauberger et al., 2020; Atzberger et al., 2015).

Currently operating low-resolution optical sensors offer a timeseries length close to 30 years, the reference length for long-term statistics according to the World Meteorological Organization (WMO, 2017). The longest and most-used timeseries for vegetation monitoring include MODIS (Moderate-resolution Imaging Spectroradiometer) with 24 years of data; the combined dataset VGT-PV-S3 offering 26 years (VGT stands for Satellite Pour l'Observation de la Terre, SPOT-VEGETATION; PV for the VGT instrument onboard its successor mission Proba-V; and S3 for the Ocean and Land Colour Instrument onboard Sentinel-3); and 42 years from AVHRR (Advanced Very High Resolution Radiometer) that was flown on multiple satellite platforms. Lastly, VIIRS (Visible Infrared Imaging Radiometer Suite) timeseries provides 12 years of data and is specifically designed to ensure continuity with the MODIS timeseries (Román et al., 2024) over an extended period. Indeed, the JPSS (Joint Polar Satellite System) program includes three operational VIIRS sensors, i.e., Suomi-NPP (Suomi National Polar-orbiting Partnership), NOAA-20 and NOAA-21 (National Oceanic and Atmospheric Administration), and the two upcoming ones (JPSS-3 and 4) planned to operate through the late 2030s.

As indicator of biomass condition, vegetation indexes (e.g., the Normalized Difference Vegetation Index, NDVI; the Enhanced Vegetation Index, EVI) or biophysical variables (e.g., the Fraction of Photosynthetically Active Radiation, FPAR) are typically used in operational crop and rangeland monitoring (Cammalleri et al., 2021; Rojas, 2021; Rembold et al., 2023; Wu et al., 2015) and yield forecasting systems (Meroni et al., 2021; Paudel et al., 2021; Mateo-Sanchis et al., 2023). In particular, FPAR is more closely linked to canopy processes, and it is a key biophysical variable for estimating vegetation productivity and monitoring terrestrial carbon (Monteith, 1972; Xiao et al., 2019). FPAR is defined as the fraction of incident photosynthetically active radiation (PAR, radiation in the 400-700 nm spectral region used by plants in photosynthesis) absorbed by the green elements of the vegetation canopy, and it is recognized by the global climate observing system (WMO et al., 2006) as an essential climate variable (ECV). Unlike vegetation indexes, which depend on spectral responses of sensor-specific bands,

illumination and observation angle and canopy background, FPAR is an inherent canopy property and can be retrieved on

observations provided by sensors with different spectral characteristics, ensuring data continuity across satellite missions.

MODIS-FPAR and VIIRS-FPAR products (Myneni, 2020; Park et al., 2018a) are retrieved with the same approach and specifically produced to guarantee the continuity of the MODIS mission (Román et al., 2024). The FPAR algorithm used for MODIS-FPAR was adjusted to the VIIRS spectral characteristics (Park et al., 2018a). The most recent VIIRS-FPAR products of Collection 2 are derived from the data of two satellites (Suomi-NPP from 2012 and NOAA-20 from 2018), deploying a cross-

60 calibration of selected reflective solar bands using MODIS Aqua as a reference to reduce the bias between the reflectances of the two satellites (NASA, 2022). These efforts ensure the continuous provision of a consistent global FPAR data set compatible with the MODIS-FPAR, as the MODIS instruments will soon be phased out. The end of the production of the science products, initially planned for August 2023, was extended up to May-April 2027 (Terra) and August 2026 (Aqua) at maximum (LAADS-DAAC, 2024). In 2022, problems had already been encountered: Aqua in April (LP-DAAC, 2022a) and Terra in October (LP-

65 DAAC, 2022b) stopped data delivery due to technical problems and orbital shift, respectively. In 2023, non-recoverable data loss events for Aqua MODIS were reported for July (LP-DAAC, 2023a). LP DAAC (The NASA Land Processes Distributed Active Archive Center) announced on 17 August 2023 that the Flight Operation Team for Terra and Aqua MODIS transitioned to Light-Out-Operations, which can result in additional data losses and larger data gaps (LP-DAAC, 2023b).

An intercalibration of the VIIRS-FPAR timeseries with that of MODIS is feasible with relatively limited efforts thanks to

70 the similarity of the sensors in terms of spatial and spectral resolution, and algorithms for FPAR retrieval. A longer timeseries with near-real-time (NRT) data would have been possible using additional sensors (i.e. VGT, PV, S3, AVHRR). However, this would require the harmonization of FPAR data produced by different algorithms and different spatial resolutions (e.g., VGT with spatial resolution of 1000 m, PV and S3 with 300 m). The AVHRR timeseries presents even greater challenges, with more than 15 sensors to consider and no FPAR product, but only NDVI at a coarse 8 km resolution (Pinzon and Tucker, 2014;

Pedelty et al., 2007). The above options present strong intercalibration challenges, which make results likely less reliable as compared to the intercalibration of MODIS and VIIRS FPAR products.

Besides the need for consistent, long-term timeseries, EWS and CYFS require high quality, continuous and updated information for effective and timely decision-making by stakeholders. For this reason, noise and cloud contamination removal is required both for the historical archive and in NRT production. For historical observations, temporal smoothing can effectively

reduce noise and cloud contamination while filling gaps in the timeseries (Goward and Huemmrich, 1992; Chen et al., 2004; Weiss et al., 2014), as data points are available before and after each observation to be smoothed. For NRT data, specific filtering methods need to be developed, to handle unbalanced data availability around recent data points (Klisch and Atzberger, 2016; Meroni et al., 2019).

Several products exist that offer high-quality timeseries of biophysical variables, such as HiQ-LAI (Yan et al., 2025) or

85 GIMMS FPAR4g (Zhao et al., 2024), but they usually lack some of the mandatory features needed by operational agricultural monitoring systems (i.e., long-term record and NRT availability). Typically, there is no guaranteed NRT data delivery into the future, nor are datasets filtered to reduce atmospheric influences. An exception is the Copernicus Land Monitoring Service that provides continuous, NRT, and filtered timeseries of biophysical variables from Proba-V and Sentinel-

3 satellites (https://land.copernicus.eu/en/products/vegetation/fraction-of-absorbed-photosynthetically-active-radiation-v1-0-300m). Nevertheless, the timeseries length offered is too short (around 12 years) for robust statistical analysis needed for anomaly computation or crop yield forecasting. In this study we fill this gap by proposing a new dataset meeting the requirements of continuous, NRT, and filtered biophysical timeseries for more than 20 years.

This paper describes a new 500 m dataset composed of two filtered and intercalibrated FPAR timeseries, one from MODIS and one from VIIRS. This dataset was produced to support the operational crop monitoring and yield forecasting activities of the Joint Research Centre of the European Commission. These include the European Mars-Crop Yield Forecasting System (M-CYFS) and the global Anomaly hotspot of Agricultural Production (ASAP, https://agricultural-production-hotspots.ec.europa.eu/) early warning system. The FPAR dataset is accompanied by associated quality layers and has a temporal resolution of 10-day, a time step often used in operational agricultural monitoring. The dataset is open and freely available in NRT through the Joint Research Centre Data Catalogue (https://data.jrc.ec.europa.eu/dataset/1aac79d8-0d68-4f1c-a40f-b6e362264e50) and on the ASAP website (https://agricultural-production-hotspots.ec.europa.eu/data/MO6_FPAR). This paper has the following specific objectives: *i*) to introduce the method used to produce a long-term archive of NRT filtered FPAR data; *ii*) to present the intercalibration performed between the filtered MODIS-FPAR and the filtered VIIRS-FPAR; *iii*) to evaluate the robustness of the FPAR filtering; and *iv*) to describe the open and free dataset and discuss its sustainability. The quality of the input FPAR products relative to ground observation is not within the scope of this study and is described elsewhere (Yan et al., 2025).

## 2 Input data

We used the MODIS-FPAR timeseries from Collection 6.1 (2000-2023) and the VIIRS-FPAR timeseries from Collection 2 (available at the time of analysis from 2018 to 2023), both with 500 m spatial resolution.

### 2.1 MODIS FPAR

The MODIS FPAR products are retrieved from a main algorithm that considers the vegetation structural type, the sun-sensor geometry, the Bidirectional Reflectance Factors (BRFs) at red and near-infrared spectral bands and their uncertainties (Knyazikhin et al., 1999; Myneni, 2020). A back-up algorithm is applied only for cases where no suitable solution is obtained from the main algorithm, and relies on the empirical relationship between NDVI and canopy FPAR (Myneni, 2020). MODIS-FPAR products were collected from the most recent Collection 6.1 through the Data Pool at the LP DAAC. From 18-02-2000 to 26-06-2002 the timeseries consists of MODIS-FPAR from Terra (MOD15A2H, 8-day composite, Myneni et al. (2021b). Since 04-07-2002, a MODIS Terra and Aqua combined product is available (MCD15A3H, 4-day composite, Myneni et al. (2021c). In case of missing MCD15A3H, we used MOD15A2H (Terra) or MYD15A2H (Aqua, Myneni et al. (2021a), if available. This occurred in 2022, when MCD15A3H data production was interrupted in April (LP-DAAC, 2022a), and in October (LP-DAAC, 2022b), while the MYD15A2H or MOD15A2H products were available (Terra in April 2022, Aqua in October 2022). For each time composite, we collected the global terrestrial coverage, composed by 286 tiles of the MODIS sinusoidal tile grid. Each

MODIS tile is provided in Hierarchical Data Format (HDF), from which we extracted three layers: Fpar_500m, FparLai_QC and FparExtra_QC. Fpar_500m layer contains for, each pixel, the maximum FPAR value among daily retrievals within the composite period (8-day for MOD15A2H and MYD15A2H or 4-day for MCD15A3H). The FparLAi_QC and FparExtra_QC layers are defined as the quality assessment (QA) products, meant for selection of reliable FPAR values. The QA products contain information about data source, detector problems, cloud and cloud shadow presence, the algorithm used to retrieve FPAR, the presence of snow or ice, the presence of aerosols, and the presence of cirrus (see Table A1 for a detiled description). Because no reference date is provided for the MODIS 4-day or 8-day FPAR products, we set their nominal date to the last day of the compositing period.

## 2.2 VIIRS FPAR

For VIIRS, we used the Collection 2 which includes VNP15A2H (Suomi-NPP satellite, Myneni (2023b) and VJ115A2H (NOAA-20 satellite, Myneni (2023a) FPAR products available at the time of analysis (30-06-2023). Collection 2, available at the time of analysis from 2018, has the same spatial (500 m) and temporal (8-day composite) resolution as the MODIS-FPAR products (MOD15A2H and MYD15A2H). The VIIRS-FPAR Collection 2 aimed at improving consistency with MODIS products by implementing a cross-calibration to limit the bias to maximum 1% for selected reflective solar bands, using MODIS Aqua as a reference (NASA, 2022; Román et al., 2024). We downloaded VIIRS-FPAR Collection 2 in HDF format from NASA's Earthdata cloud. VIIRS-FPAR products have a data structure similar to MODIS-FPAR products with three layers provided: Fpar_500m, FparLai_QC and FparExtra_QC. However, some differences are present in the QA products. Most notably, the logic of the cloud coverage information was changed: for VIIRS four levels of cloud probability are provided instead of the *Internal cloud mask* and the *Cloud state* provided by the MODIS QA products. Similarly, the aerosol presence is classified in four levels instead of the single flag provided for MODIS. TableA2 describes in more detail the VIIRS QA layer contents. Because no reference date is provided for the VIIRS products, we set this date to the last and before-last day of the compositing period (the 8th day for VNP15A2H and the 7th day for VJ115A2H). Different dates are assigned because our smoothing implementation accepts one value per day.

## 3 Study area and ancillary data

Our dataset targets the global extent covered by the MODIS and VIIRS original products, approximately from 75° North to 56° South and from 180° West to 180° East. It uses a temporal step of 10-day (i.e. dekad), typically employed in agronomic analysis. A dekad is a nearly 10-day period covering each month with 3 dekads (day 1–10, 11–20, and 21–last day of the month) and the calendar year with 36 dakads. To focus the analysis of the dataset on the vegetation growing cycle, statistics were extracted on the average growing season period defined per pixel using the phenology layer of the ASAP system. ASAP defines start and end of a fixed growing season at pixel level based on thresholds on the green-up and decay phases (Rembold et al., 2015). Because FPAR is produced also for pixels with a presence of permanent or seasonal water, we masked such pixels out. To mask water pixel, we used a land/water mask derived from the MOD44W Collection 6, a global map of surface water

at 250 m spatial resolution in the standard MODIS sinusoidal grid (Carroll et al., 2017). This mask was aggregated to 500m spatial resolution by labelling a pixel as land only if no 250m water-pixel was included. From the remaining, non-watered pixels, we selected all vegetated pixels accordingly to MCD12Q1 biome map (Friedl and Sulla-Menashe, 2019) which cover crops, shrubs, savanna, and forests. Since the main scope of our dataset is agricultural monitoring, we used two further masks (cropland and rangeland) from the JRC-ASAP system (Fritz et al., 2024) to analyse our results.

## 4  Methods

The main steps to produce our combined filtered FPAR dataset are shown in Fig.1. In this paper, we adopt the definitions according to Sedano et al. (2014): smoothing refers to the interpolation over a time span when observations are available before and after each data point, while filtering refers to the estimation of near-real-time (NRT) data using only past observations. The processing steps regarding smoothing are described in Sections 4.1.1 and 4.1.2, while filtering in Section 4.1.3. Methods used to assess the quality of the filtering are described in Section 4.2, while the comparison and alignment between VIIRS-FPAR filtered data and MODIS-FPAR filtered data is described in Section 4.3.

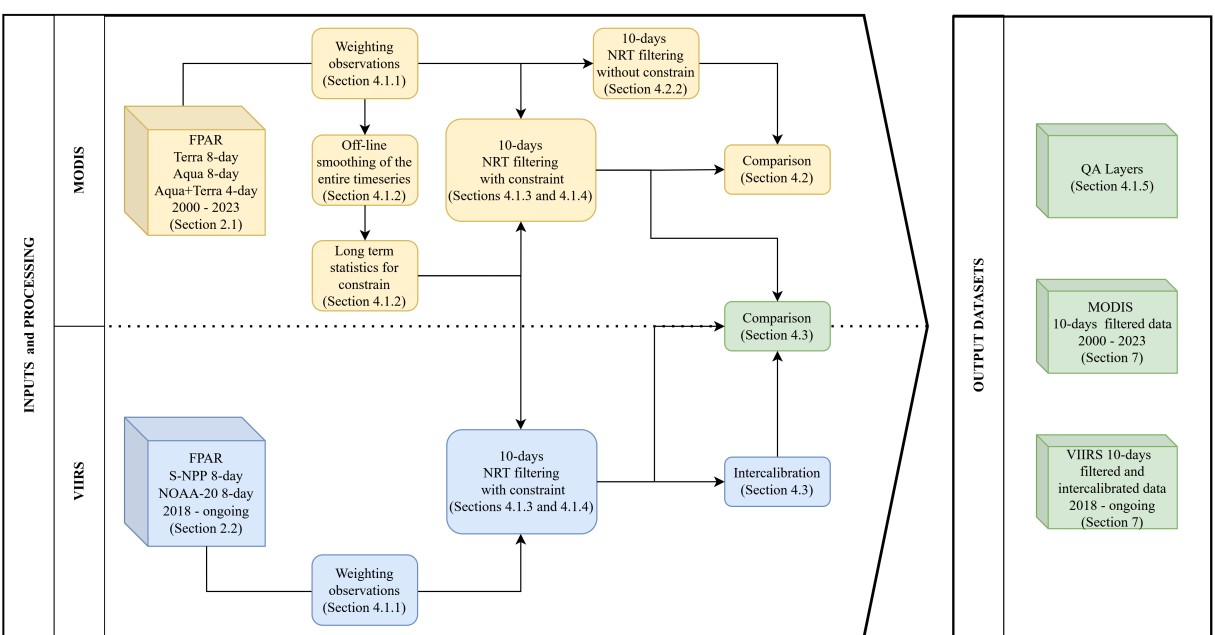

**Figure 1.** Workflow for generating the NRT FPAR dataset from the input FPAR data and the main comparison analysis provided in this paper. The reference to section of the paper describing the associated building block is reported in brackets. The blocks associated to MODIS data are in yellow, those associated to VIIRS data in blue, and those to the outputs in green.

## 4.1 FPAR smoothing and NRT filtering

Our approach builds on the previous method for developing a NRT operational MODIS NDVI product (Klisch and Atzberger, 2016; Meroni et al., 2019) based on the Whittaker smoother (WS (Eilers, 2003; Atzberger and Eilers, 2011a, b). In our implementation of the WS, the FPAR observations are weighted according to the QA products (Section 4.1.1), while all the available observations are used. Compared to the previous implementations, we revised the weights assigned to MODIS data and introduced new weights for VIIRS data. The WS achieves a balance between the fidelity to the original data and the roughness of the smoothed curve (i.e., the second-order differences) by tuning its smoother parameter $\lambda$: a larger $\lambda$ results in smoother results that align less with original data. In Section 4.1.2 we describe the tuning of the parameter $\lambda$ and the smoothing of the whole timeseries (i.e., off-line smoothing) for the computation of the long term statistics. Compared to the previous implementations, we revised the parameter $\lambda$ and introduced an iterative upper envelope fit approach to minimize noise from undetected cloudy observations (Chen et al., 2004). The operational NRT filtering is described in Section 4.1.3 while its adaptation in presence of sparse observations is presented in Section 4.1.4. Finally, in Section 4.1.5 we describe the quality products associated with the NRT filtering. Compared to the previous implementation, this study modified the adaptation strategy and perfected the system of the quality layers.

### 4.1.1 Weighting observations

FPAR data are weighted according to the MODIS and VIIRS QA products. The attribution of weights to specific combinations of quality indicators was based on the visual inspection of a large amount of observations, aimed to understand the relationship between specific quality indicators and data quality. This process led to the definition of three different weights associated to specific quality indicators (Table 1). The weighting scheme was developed by visually inspecting 117 pixel-based FPAR timeseries between 2000 and 2021. Those 117 samples were obtained from a stratified random sampling over the MCD12Q1 biome map. We considered 70 samples from the biomes cropland-grasslands, broadleaf cropland, and savanna, comprising 35 in Europe and 35 in Africa. The other 47 samples were selected from all eight available biomes for the whole globe (Fig. B1, Table B1). The reliability and consistency of each timeseries was visually assessed and, with focus on short timescale consistency, assuming that no negative spikes (i.e. rapid drop and rise of FPAR value) should occur. In assigning the weights, we followed the recommendations of Myneni (2020) suggesting that the reliability of an observation is primarily driven by the algorithm used. We assigned the highest weight (100%) to high quality -HQ- observations (i.e., cloud- and snow-free observations) from the main algorithm. We observed that in various cases HQ observations were too few, leading to poor WS performance; however backup algorithm provided consistent value. Therefore we retained the values from the backup algorithm, assigning them a weight of 50%. In addition, we found that cloud or snow contaminated observations frequently had plausible FPAR values. In the attempt to increase the number of usable observations, we assigned a small weight of 20% to pixels flagged as contaminated by either the main or the back-up algorithm. Possible drawbacks of retaining cloudy observations are mitigated by the upper-envelope adaptation (described in Section 4.1.2) that is designed to downweight cloudy and negatively biased observations. The visual inspection of the timeseries also revealed that *aerosol* flags occurred very frequently and, in most cases, without

detectable impact on FPAR observations. Therefore, we decided to downweight to 20% only those observations flagged as *aerosol* when MODIS-FPAR data is retrieved with the backup algorithm, while for VIIRS-FPAR data downweighting to 20% occurred when both *Climatology* and *High* flags were marked in the QA products, as suggested by Lyapustin et al. (2021). Finally, we observed a number of cases with exceptional (and unrealistic as compared to values immediately before and after) FPAR values of 100%, negatively impacting the smoothing results. In principle, both the main and backup algorithm can result in values large as 100% (FPAR NASA Science Team, personal communication), but in the majority of cases we observed those values when the backup algorithm was used. As consequence, we weighted these erroneously high values as 0%, together with those flagged with *Fill value* (no realistic observation) and *Dead detector* (physical error at sensor) in the QA products.

**Table 1.** Weights proposed for different observation conditions. For the *Gap-filling procedure applied* refers to Section 4.1.4

| Weight in % | Description |
|---|---|
| 100 | Main algorithm, no clouds and no snow present |
| 50 | Backup algorithm, no clouds, snow, or aerosol present |
| 30 | Gap-filling procedure applied |
| 20 | Clouds or snow present |
| 20 | Aerosol present and backup algorithm |
| 0 | FPAR = 100 and backup algorithm |
| 0 | Dead detector |
| 0 | Fill value |

### 4.1.2 Off-line smoothing

Off-line smoothing refers here to a smoothing that is performed retrospectively on the full historical timeseries and not in NRT. This smoothing is performed once and only to extract the pixel-based statistics that serve the NRT filtering. We performed the off-line smoothing on MODIS data over the period 01-01-2003 to 31-12-2021 that can be considered representative for normal MODIS operations, when both Terra and Aqua satellites were available and fully operational (no sensor issues). Only MODIS data were considered as they offer a significantly longer timeseries compared to the VIIRS timeseries, and as such, are better suited for statistical analysis. Only pixels with at least 80 HQ observations in the entire timeseries were retained (flagged as no data otherwise). We set the WS smoother parameter $\lambda$ value to 3000 following previous analysis (Klisch and Atzberger, 2016; Atzberger et al., 2015) and the visual inspection of fitted curves produced with different $\lambda$ values. The data used for the smoothing were weighted as described in Section 4.1.1. In addition, the possible residual presence of cloudy observations (downweighted or fully undetected by the quality flags) was suppressed by iteratively applying the smoothing to fit the upper envelope of the FPAR temporal trajectory (Chen et al., 2004; Beck et al., 2006). This approach was chosen for its flexibility in capturing vegetation signal changes and its successful validation in previous studies (Meroni et al., 2014). From WS's daily output, FPAR rasters are stored for one moment within the 10-day (dekad) period. This moment was fixed to the

5th day of the dekad corresponding to 5th, 15th, 25th of each month. From the complete timeseries of smoothed 10-day rasters, two classes of long term statistics are computed per pixel: long-term per-dekad average (LTA) and dekad-to-dekad variations (i.e., minimum, average and maximum difference between every two consecutive dekads of the year, computed over the full historical timeseries).

### 4.1.3   NRT filtering

The NRT filtering method consists of a modified version of the WS applied at the end of every 10-day period from 01-07-2023 onwards. Prior to that date, FPAR data were filtered with NRT method in hindcasting (i.e., by simulating the lack of data later in time than the dekad to be filtered). Indeed, while the off-line smoothing consider all the dekad at once, with hindcasting we simulated the operational constraints to obtain a consistent dataset computed in NRT mode. Hindcasts were done for MODIS from 21-05-2000 onwards, and for VIIRS since the start of Collection 2, at time of analysis (01-01-2018).

The NRT filtering principle is the same as described in Section 4.1.2, while its application is slightly different. In NRT, we use the FPAR observations within a time window of 190 days before the day of the filtering (TWL, Temporal Window Length), and constrain the output filtered values under certain conditions (4.1.4). Our filtering estimates the FPAR value for the latest 10-day period, C0, and the four previous 10-day periods. In this way, as time passes, the same FPAR value is first produced (consolidation stage 0, C0) and then updated four times (consolidation stages 1 to 4: C1, C2, C3 and C4), before reaching a

final consolidation stage CF, 90 days after its first C0 estimation. The filtered values between C4 and CF (e.g., C5-C8) are not stored, as the impact of the update is minimal.

### 4.1.4   NRT filtering with sparse observations

When HQ observations in the latest 40 days of the TWL are sparse, WS estimates for earlier stages, and notably those close to C0, are prone to provide unrealistic FPAR values because of extrapolation effects, as only past observations are available.

To limit this effect Klisch and Atzberger (2016) introduced a constraint to the filtered FPAR values. Here, this is implemented separately for two different cases: *i*) at least one HQ observation in the last 40 days from the date of smoothing, *ii*) no HQ observations in that period. In case *i*) an anchor point is defined as the latest estimated stage which is followed by at least one HQ observation (e.g., C3 in Fig.2 - panel *b*)). For the subsequent stage (e.g., C2), we accept filtered FPAR value if its variation from the anchor point remains within the minimum-maximum range, derived from dekad-to-dekad variation obtained from

the off-line smoothing procedure (Section 4.1.2). If the filtered value exceeds this range, it is truncated to the correspondent boundary value. The new filtered (and potentially constrained) stage (e.g., C2) then becomes the anchor point for the next iteration until the C0 value is calculated. In case *ii*), an initial gap-filling procedure is used. The first anchor point is set to the C4 data point from the previous date of filtering (e.g., star point in Fig.2 - panel *c*). Then, to gap-fill the missing observation for the next 10-day period, a synthetic FPAR value is computed based on LT statistics (with given weight of 30%). However,

instead of using the LT average directly (that may be far larger or smaller in case the current season is better or worse than the average season) the synthetic value is obtained by adding to the anchor point FPAR value the corresponding average dekad-to-dekad variation. This new point becomes the anchor for the next iteration. The process is repeated up to computing five

FPAR replacement values (C4 to C0). The filtering and the constraint mechanisms are then applied, as in (*i*). In the exceptional case of less than five HQ observation in the TWL, missing values are replaced directly with corresponding LTA values, and
the filtering and constraint mechanisms are applied as in, as in (*i*). With the filtering procedure, we obtained two timeseries of filtered data: one for MODIS-FPAR and one for VIIRS-FPAR.

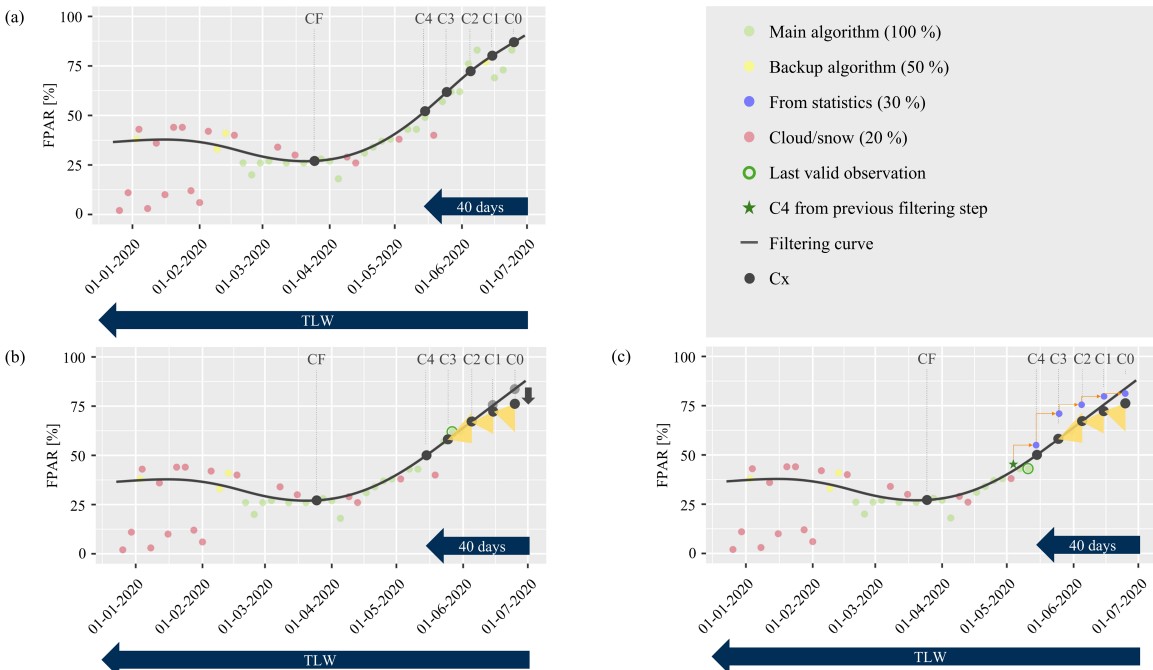

**Figure 2.** Panel *a*) shows an example of the NRT filtering approach with enough HQ observations in the latest 40 days and no constraint mechanism applied. Panel *b*) displays an example of applying the constraint mechanism that modifies the values of C0 and C1 as their first SW filtering estimation is out of the admissible value range (i.e., yellow triangles). Panel *c*) shows an example of applying the gap-filling and constraint mechanism. The blue dots are the gap-filled values as from the LTA. After the gap-filling, the filtering proceeds with filtering and constraint mechanism.

### 4.1.5 NRT quality layers

We keep trace of our filtering mechanism path (e.g. the application of gap-filling procedure) in a status map (SMP), produced for each dekad and consolidation stage at pixel level (Section 7). Together with the SMP we produce four additional QA layers:
number of HQ observations between stages C4 and C0 (NWM), average weight of observations between stages C4 and C0 (QWM), number of days from the last HQ observation to the last day of the 10-day window (NLM), and weight of last available observation (QLM). NWM, QWM, NLM, and QLM refer to quality and availability of observations within the TLW. Indeed, the lower the number of HQ observations available (NWM and NLM) and the smaller the weight of the observations used

(QWM and QLM), the less reliable are the FPAR estimates for the unconsolidated stages. The QA layers are stored once with the reference date of C0 and are valid for all stages produced at the specific date.

## 4.2 FPAR filtering assessment

To assess the quality of the filtering we: *i*) evaluated its robustness by comparing, for each single dekad, the FPAR value of the unconsolidated stages (i.e., C0 to C4) with the value of the consolidated stage (i.e., CF), assumed to be the best FPAR estimate (Meroni et al., 2019); *ii*) evaluated the utility of the constraint mechanism (Section 4.1.3) by an ablation study. In both cases, the assessment was performed on a subsample of the FPAR global rasters, following the approach described Meroni et al. (2019). Our sample was obtained by spatially subsampling the global rasters by selecting the central pixel within a not-overlapping window of 41x41 FPAR pixels; this approach reduced computational time but still captured global vegetation patterns (Toté et al., 2017). As a result, the sample is composed of timeseries from 324,004 pixels. All the stages were computed using only the filtered MODIS-FPAR timeseries as it offers a significantly longer timeseries compared to the VIIRS timeseries, and thus provides more robust statistics.

### 4.2.1 Filtering robustness

To assess the robustness of the filtering, we compared the FPAR stages from C4 to C0 against CF, by dekad. We computed the mean absolute error (MAE) and the mean error (ME) for each pixel and each MODIS dekad between 01-05-2003 and 31-12-2021. We expressed them as Cx (consolidation stage x) error compared to CF, with the notation *_Stage_Cx*, following Eq. (1) and Eq. (2. )

$$MAE\_Stage\_Cx = \sum_{y}\sum_{i} \frac{|FPAR\_Cx_{yi} - FPAR\_CF_{yi}|}{N} \tag{1}$$

$$ME\_Stage\_Cx = \sum_{y}\sum_{i} \frac{(FPAR\_Cx_{yi} - FPAR\_CF_{yi})}{N} \tag{2}$$

*Cx* is the consolidation stage $x$ ($x = 0, \ldots, 4$), *CF* is the final and reference consolidated stage, $y$ is the year (19 years in total), $i$ is the dekad, and $N$ the total number of samples. The metrics were temporally aggregated over the average growing season of each pixel and spatially averaged over the three strata: vegetated, cropland, and rangeland (Section 3).

### 4.2.2 Ablation study

To assess whether the constraint mechanism results in more accurate NRT FPAR estimates, we performed an ablation study. We generated a non-constrained MODIS-FPAR filtered timeseries using the same settings as in Section 4.1.3, but without applying the constraint mechanism and we then computed MAE_Stage_Cx and ME_Stage_Cx (see Section 4.2.1) for the non-constrained timeseries. Finally, we compared these error metrics to those obtained from the constrained-timeseries. We used data from 01-01-2003 to 31-12-2021, from MODIS only, for the same reasons outlined in Section 4.2.

### 4.3 Intercalibration of FPAR filtered timeseries

With the objective of combining the VIIRS and MODIS FPAR filtered timeseries, we first visually compared them and found
spatial and temporal differences. As these differences may affect operational monitoring and forecasting activities, we decided
to systematically assess the presence of bias and to intercalibrate the filtered VIIRS-FPAR timeseries on the longer filtered
MODIS-FPAR timeseries. Such analysis was done over the 5 years of overalpping data available at the time of analysis, between
01-07-2018 and 30-06-2023. After initial test with several intercalibration methods (Ceccherini et al., 2013; Cammalleri et al.,
2019; Gudmundsson et al., 2012) we opted to apply the mean difference MD correction. MD correction has the advantages that
can be applied using a short overlap period as MODIS-FPAR Collection 6.1 and VIIRS-FPAR Collection 2 and can account for
temporal and spatial differences, where applied per-dekad and per-pixel. We computed the global rasters of the dekadal MD
between the filtered FPAR timeseries of MODIS and VIIRS for each consolidation stage. Each filtered FPAR value derived
from filtered VIIRS-FPAR timeseries was then corrected by adding its corresponding dekadal MD value, pixel-specific and
consolidation stage-specific.

### 4.3.1 Intercalibration assessment

To assess the differences between the same stage of the filtered and intercalibrated VIIRS FPAR timeseries and the filtered
MODIS timeseries we used again MAE and ME but with the notation _*Sensor* to mark the difference in the domain of applica-
tion and coverage period as compared to Section 4.2.1. MAE_Sensor and ME_Sensor were calculated for each pixel and each
dekad between 01-07-2018 and 30-06-2023, following Eq. (3) and Eq. (4).

$$MAE\_Sensor\_Cx = \sum_{y}\sum_{i} \frac{(|FPAR\_VIIRS\_Cx_{yi} - FPAR\_MODIS\_Cx_{yi}|}{N} \tag{3}$$

$$ME\_Sensor\_Cx = \sum_{y}\sum_{i} \frac{((FPAR\_VIIRS\_Cx_{yi} - FPAR\_MODIS\_Cx_{yi})}{N} \tag{4}$$

*FPAR_VIIRS* is the filtered and intercalibrated FPAR timeseries of VIIRS, *FPAR_MODIS* is the filtered timeseries of
MODIS, *Cx* is the consolidation stage *x* ($x$ = 0, ..., 4, F). MAE_Sensor and ME_Sensor were temporally and spatially ag-
gregated as those of Eq. (1) and Eq. (2.)

## 5  Results

### 5.1  NRT filtered FPAR timeseries

We produced two global timeseries of filtered FPAR, one based on MODIS-FPAR filtered data and one on the intercalibrated
VIIRS-FPAR filtered data. A total of 16 global rasters were produced every 10 days for both series (from 20-08-2000 to 31-
12-2023 for MODIS, since 01-01-2018 for VIIRS). Each set of 16 rasters was composed by the six FPAR consolidation stages

(i.e., C0, C1, C2, C3, C4, CF), the associated SMP rasters (6 rasters), and the four QA rasters (NWM, QWM, NLM, QLM). Fig. 3a shows an example of the global raster from the intercalibrated VIIRS-FPAR timeseries for dekad 13 of 2023 (01-05-2023 to 10-05-2023). It nicely shows the gradient of vegetation vigour in Europe, with FPAR values around 80% in the West and between 40 % to 60 % in the East, and very low FPAR values for the Iberian Peninsula due to the severe drought conditions that spring (EC-JRC, 2023). Figure 3b illustrates the SMP for the same dekad, indicating the quality of the information provided. For the largest share of vegetated land the FPAR values were obtained with favourable filtering conditions with no need for constraint or gap-filling procedures. In contrast, constraint and gap-filled procedures were used where prolonged periods of snow or clouds occurred (orange or red colours), as at higher latitude, or at elevated altitude (e.g. the Alps) and over tropical forests. The corresponding QA layers are presented in Fig. C1a–C1d.

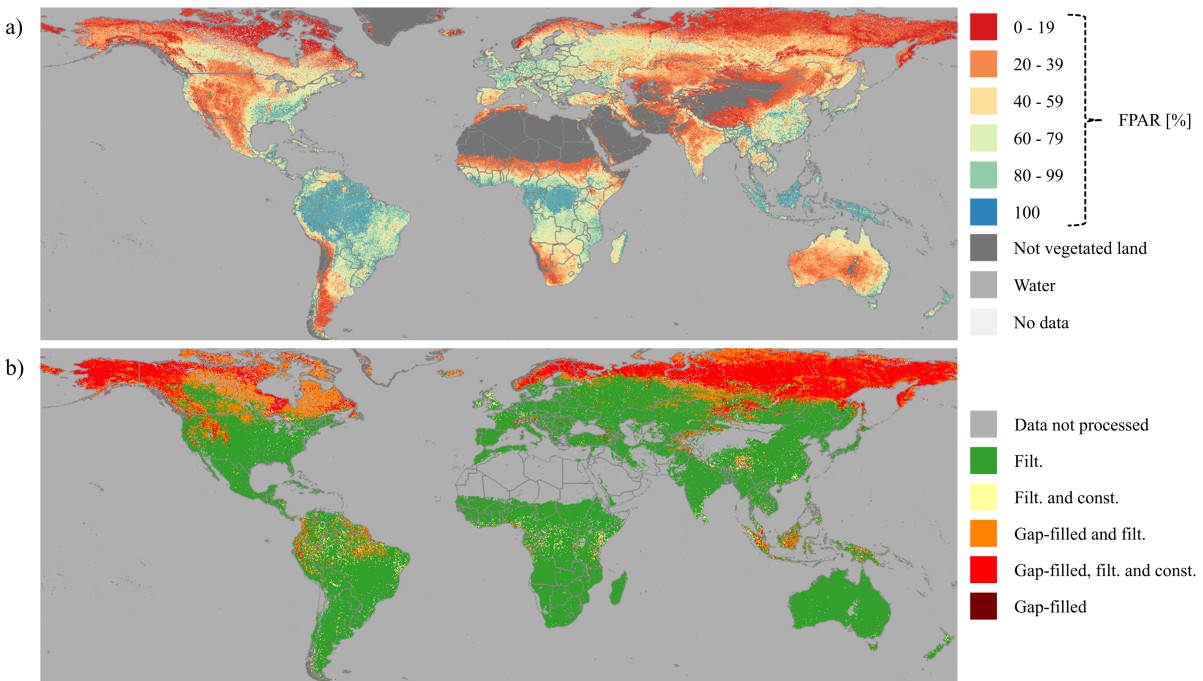

**Figure 3.** Global maps of intercalibrated VIIRS-FPAR filtered for dekad 13 of 2023 (period 1-10 May 2023). Panel a) displays the filtered FPAR values, and panel b) the SMP status map describing which operations were performed to compute the final FPAR value. The abbreviations of *Filt.* and *const.* stand for *filtering* and *constrained*, respectively.

## 5.2 NRT filtering robustness

We assessed the robustness of the NRT filtering by evaluating its accuracy in predicting CF value during the growing season. A preliminary visual assessment of the timeseries for each consolidation stage at the selected sample points showed convergence toward the CF, as illustrated in Fig. 4. For example, in early 2022, FPAR was strongly overestimated for C0, although values were still realistic and close to the LTA. From the second estimation (C1) onwards, FPAR values moved significantly closer to

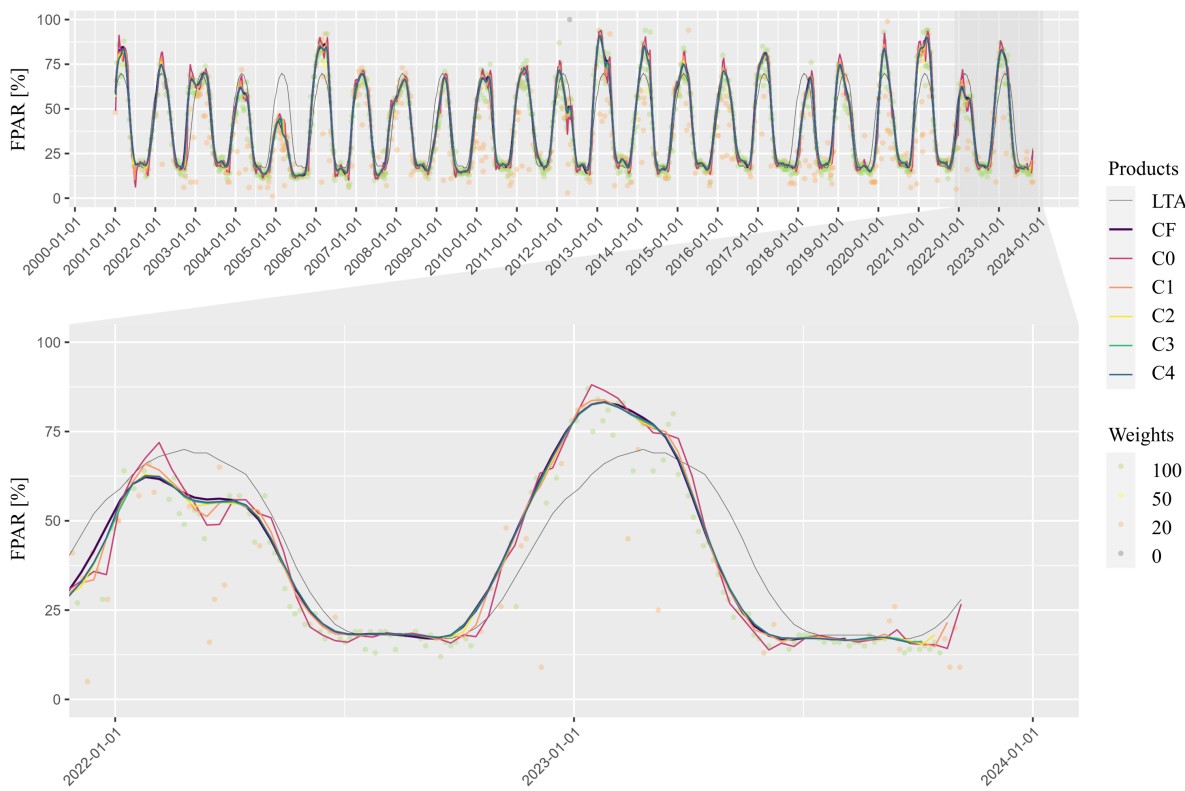

**Figure 4.** Temporal profile of MODIS-FPAR filtered data for an arable land pixel in Portugal for the period 01-01-2001 to 31-10-2023. CF, C4, C3, C2, C1, C0 represent the consolidation stages and LTA is the average computed over the CF timeseries. The bottom panel provides a detailed view of the 2022-2023 data. Dots represent the non-filtered FPAR values as from MOD15A2H.061 and MCD15A3H.061, coloured according to the weight assigned.

the final CF. Later, in March 2022, C0 underestimated the CF due to persistent low-quality observations (weighted at 20 %), whereas the C1 estimation, incorporating new HQ data, was already very close to the CF. Despite our filtering has a conservative approach relying on historic information (e.g., constraint mechanism), it effectively estimates FPAR values that deviate from the average. This is particularly evident in the 2022-2023 agricultural season, when green-up occurred much earlier than the average (Fig. 4). Fig. 5 presents a quantitative evaluation, showing the FPAR differences between all consolidation stages

and CF. The density distributions converge toward 0 % error as the consolidation progresses, indicating improved FPAR estimation with more observations. The distribution of relative errors indicates that the C2 estimation already provides a good approximation of CF, with the distribution closely resembling that of C4.

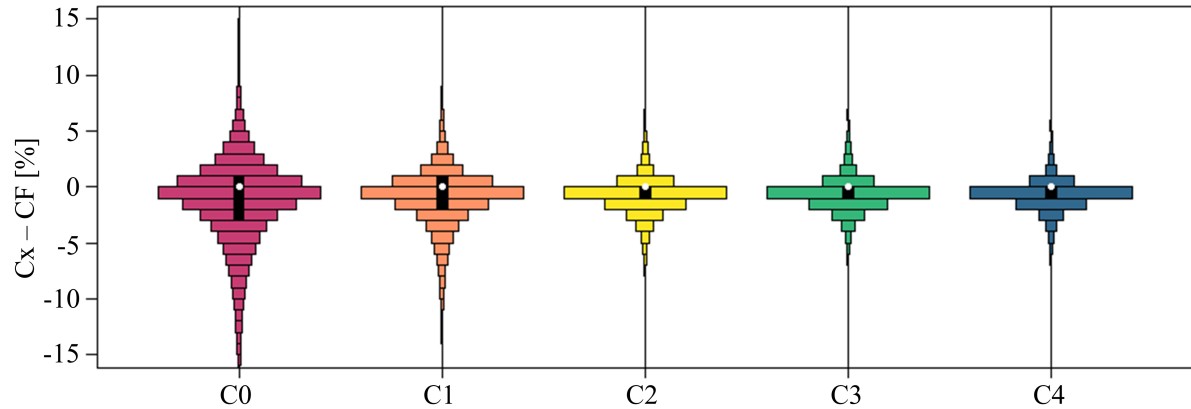

**Figure 5.** Violin plots of FPAR differences computed between the MODIS-FPAR values of each consolidation stage (C0, C1, C2, C3, C4) and the correspondent final stage (CF), for the global FPAR raster of dekad from 01-05-2023 to 10-05-2023. The values are calculated considering all vegetated pixels.

## 5.3 Ablation study results

We evaluated the utility of the constraint approach as described in Section 4.2.2. We averaged MAE for each consolidation stage, for all vegetated pixels, over all the growing seasons in the 19 available years of the MODIS-FPAR timeseries. We computed two error metrics with and without constrained filtering, which were compared stage by stage. Results (Fig. D1) showed similar spatial patterns, with the highest errors in northern latitudes and in tropical regions. Constrained filtering significantly reduced MAE, in particular for the C0 stage. High MAE in these regions was mainly due to low-quality FPAR observations (e.g., persistent cloud or snow coverage), but the constraints prevented unrealistic spikes and frequent drops in FPAR values. ME analysis for C0 confirmed that unconstrained filtering predominantly led to negative ME values, while constrained filtering resulted in slightly positive ME values. This indicated that constrained filtering at C0 mildly overestimated CF, whereas unconstrained filtering underestimated it more significantly. As expected, this effect diminished in later consolidation stages as more data became available, reducing MAE. These results demonstrate the added value of the constraint mechanism and justify its operational use.

## 5.4 Evaluation of the MODIS-VIIRS intercalibration

We compared the MODIS-FPAR and VIIRS-FPAR filtered timeseries through cumulative distribution functions (CDFs) of MAE_Sensor and ME_Sensor for each consolidation stage, with and without the MD correction. Without MD correction, the mean ME_Sensor was already very small, around -1 % across all consolidation stages, indicating a slight underestimation of MODIS-FPAR filtered values by VIIRS-FPAR filtered values. This agrees with the findings of Román et al. (2024) that examined the continuity between row MODIS-FPAR Collection 6.1 and from VIIRS-FPAR Collection, found that VIIRS slightly underestimated MODIS FPAR, and suggested the two FPAR products could be used interchangeably. The analysis

of MAE_Sensor revealed larger dispersion between the two timeseries, with most (> 70 %) pixels exhibiting a MAE_Sensor below 5 % (Rangeland and Vegetation) and 6 % (Cropland) across the majority of consolidation stages. For all land cover types considered, MAE_Sensor decreased as the consolidation stage progressed, as earlier consolidation stages (e.g., C0) are more sensitive to cloud and snow screening differences (Section 2). The MD correction reduced discrepancies across all consolidation stages (Fig. 6, continuous lines) as shown by the CDFs for the intercalibrated timeseries. This indicates that the best agreement between the two timeseries as the filtered FPAR values stabilized and approached true values (i.e., CF) with around 50% of CF pixels exhibiting a MAE_Sensor error lower than 2% for the rangeland and vegetation pixels, while slitghly higher (2.5 %) for cropland pixels. When the geographical distribution of MAE_Sensor was considered (Fig.C2) we observed consistent patterns across all consolidation stages. Larger MAE_Sensor between uncalibrated timeseries was found in areas with more persistent cloud cover (i.e. the tropics and areas at high latitude) as shown by Fig.C2a while MD correction (Fig.C2b) effectively reduced MAE_Sensor, especially at high and medium latitudes.

## 6 Example of applying the intercalibrated FPAR series for crop monitoring

We illustrate the potential use and advantages of the intercalibrated timeseries with an example application. The computation of FPAR anomalies (i.e. deviation of current values from historical average, Section 1) is a standard approach in NRT monitoring of crop biomass. Negative anomalies typically indicate biomass deficit, while positive anomalies a surplus with respect to normal. Fig. 7 showcases an anomaly assessment for arable land pixels for the county of Oise, France. We generated three relative FPAR anomaly maps for the period 1-10 May (dekad 13) 2023, using as reference the average FPAR computed over the timeseries (2002-2023) of CF stage from MODIS-FPAR data. The first anomaly map (Fig. 7*a*) is produced from the original VIIRS-FPAR data (reference doy 121, original 8-day composite VNP15A2H.002_Fpar_doy2023121000000); the second (Fig. 7*b*) is produced using C0 stage from the intercalibrated VIIRS-FPAR filtered data, the third (7-*c*)) using the CF stage from the same timeseries as in panel *b*). As CF stage represents the best FPAR estimation, we consider the resulting anomaly map as the truth.

The original VIIRS-FPAR data is limited by cloud coverage, misses both positive and negative anomalies over large areas. In contrast, our filtering approach demonstrates a clear advantage already at the C0 stage, where the entire area exhibits a consistent FPAR anomaly closely matching the CF-based anomaly. This demonstrates our method's reliability in estimating FPAR values, even with sparse and cloud-contaminated observations, when missing observations would normally be flagged as unavailable, leading to uninformed interpretations of vegetation status. Additionally, since EWS and CYFS workflows summarize data at the administrative unit level (e.g., average anomaly or FPAR value), aggregating only cloud-free areas may result in FPAR values that do not accurately reflect real conditions. Our proposed approach overcomes these limitations by providing timely, reliable data for EWS and CYFS, ensuring consistent updates for stakeholders. By estimating FPAR values despite sparse or cloud-contaminated data, it supports more accurate and informed decision-making process.

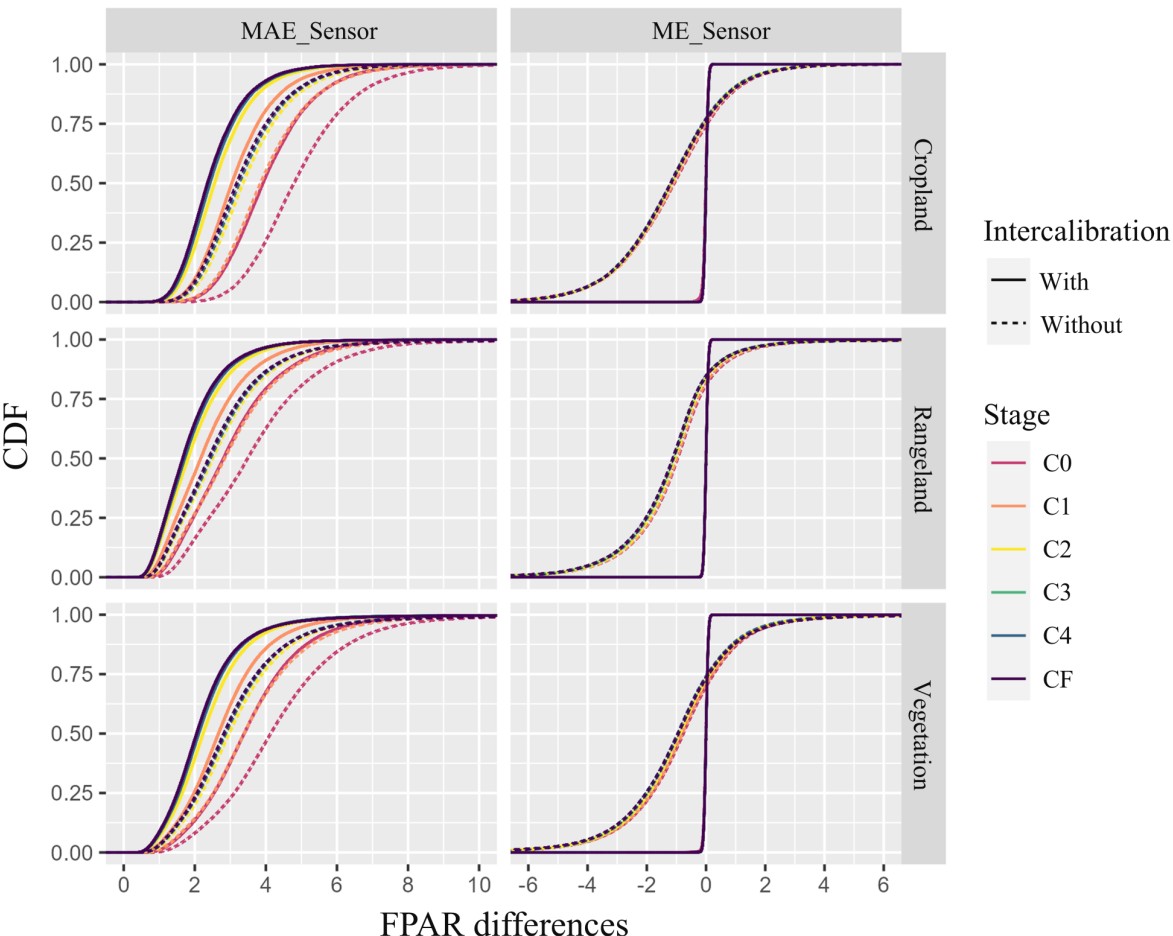

**Figure 6.** MAE_Sensor and ME_Sensor computed between the MODIS-FPAR filtered data and *i*) the VIIRS-FPAR filtered data (represented with dotted line) or *ii*) the intercalibrated VIIRS-FPAR filtered data (represented with continuous line). The analysis is provided for each consolidation stage for all global vegetated pixels, for the respective land cover classes, during the five overlapping growing seasons between MODIS-FPAR data and VIIRS-FPAR data (01-09-2018 to 31-08-2023). The x-axis shows the relative difference of FPAR values between the filtered timeseries, while the y-axis indicates the CDF.

## 7 Dataset description

We publicly released two timeseries that updated every 10 days: 1) 10-day filtered MODIS-FPAR global raster from 21-08-
395 2000 to 31-12-2023; 2) 10-days intercalibrated VIIRS-FPAR filtered global rasters from 01-07-2018 to present. Both datasets are provided together with associated QA rasters. For the MODIS dataset we provide for each global 10-day FPAR raster the CF stage and the associated SMP, the other consolidation layers could be made available upon request to the functional email JRC-ASAP@ec.europa.eu. For the VIIRS dataset we provide the following set of rasters: the six consolidation stages (C0, C1,

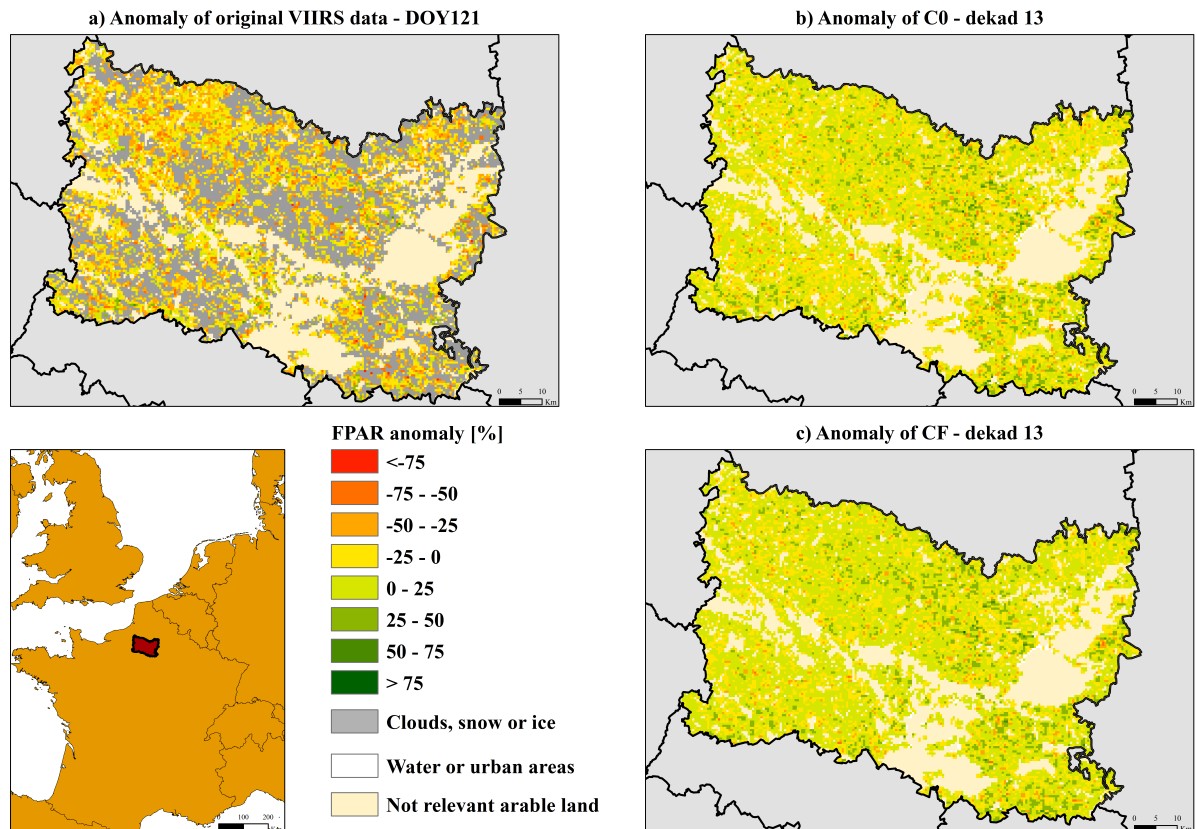

**Figure 7.** Three different maps of relative FPAR anomalies for 1-10 May (dekad 13) 2023, for the county of Oise, an important agricultural region in northern France. The historic reference used to calculate the three anomalies is the multi-year CF average of dekad 13 using the full MODIS-FPAR filtered timeseries. Panel *a*) shows the relative anomaly map obtained using the original 8-day composite VIIRS-FPAR data (doy 121 of 2023). Panel *b*) shows the relative anomaly map using stage C0 from intercalibrated VIIRS-FPAR filtered data from dekad 13 of 2023. For panel *c* the relative anomaly map is obtained using the stage CF from the intercalibrated VIIRS-FPAR filtered data from dekad 13 of 2023. Pixels masked and labelled as *Not relevant arable land* have an arable land cover less than 10%, according to Corine Land Cover 2018 (LMS, 2018)

C2, C3, C4, CF), the associated SMP for each consolidation stage, and the four QA layers (i.e., NWM, QWM, NLM, QLM). All rasters are provided in geographic coordinates (EPSG 4326) with a spatial resolution of approximately 500 m (0.004464°). Water pixels are masked using the land/water mask, as from Table 3. The output format is a compressed GeoTIFF. All outputs are produced in NRT time mode (actual or hindcast). FPAR values are scaled to 8-bits in the same way as to the original files (Myneni, 2020; Park et al., 2018a). The FPAR rasters have a dimensionless physical unit (ratio), 8-bit data type, and the valid data range is rescaled to 0-100. Flag values are reported in Table 3. Each NRT global raster and its quality layers are released two days after the end of the dekad (e.g., for the dekad 1-10 May, the releasing date would be 12 May). Apart from the quality

control internal to the filtering operational process (e.g., check for missing tiles) at each release the products are randomly sampled and visually inspected to control for visible artefacts.

**Table 2.** Quality information stored during NRT filtering per dekadal raster, SMP is provided for each stage, the other indicators are provided for C0 only, all layers have data format of 8-bit unsigned integers.

| Parameter | Description | Units | Flag values | Valid range |
|---|---|---|---|---|
| SMP | Status map indicating the filtering condition | - | 1. Data not processed (water, other land) <br> 2. Filtered <br> 3. Filtered and constrained <br> 4. Gap-filled and filtered <br> 5. Gap-filled, filtered and constrained <br> 6. Gap-filled (no input files at all available) | - |
| NWM | Number of HQ observations between stages C4 and C0 | - | 251, 254, 255 | 0-13 |
| QWM | Average weight of observations between stages C4 and C0 | % | 251, 254, 255 | 0-100 |
| NLM | Number of days from the last HQ observation to the last day of the temporal window | Days | 250, 251, 254, 255 | 0-190 |
| QLM | Weight of last available observation | % | 251, 254, 255 | 0-100 |

**Table 3.** Flag values of the output rasters

| Flag value | Description |
|---|---|
| 250 | (NLM only) no HQ observations |
| 251 | non-vegetated land |
| 254 | water |
| 255 | no data (e.g. too few observations) |

## 8    Recommended use of the timeseries

In view of an operational use of our dataset, we suggest the approach depicted in Fig.8 to select the relevant stages and
timeseries for NRT analysis

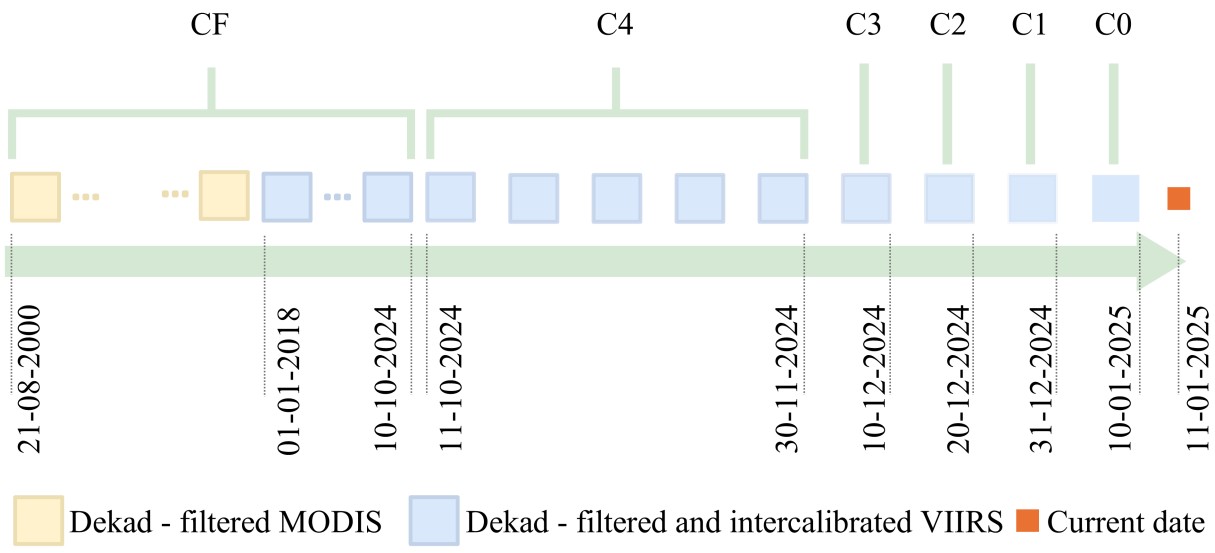

**Figure 8.** Example of suggested operational timeseries as would be at the date of 11-01-2025

Suppose that the current date is the 11-01-2025 the operational timeseries First, in an attempt to use the highest data quality
we recommand to use CF outputs from the beginning of the timeseries (dekad 24 of 2000, covering dates from 21-08-2000
to 31-08-2000) until the last produce CF stage, i.e. 90 days ago (dekad 28 of 2024, from 01-10-2024 to 10-10-2024). From
that date up to 40 days ago (30-11-2024) we would rely on C4 stage, having the highest quality among the NRT consolidation
stages. The last four dekads we would use C3 (dekad 34 of 2024, from 01-12-2024 to 10-12-2024), C2 (dekad 35 of 2024, from
11-12-2024 to 20-12-2024), C1 (dekad 36 of 2024, from 21-12-2024 to 31-11-2024) and C0 (dekad 1 of 2025, from 01-01-
2025 to 10-01-2025) stages. Second, during the overlap period between our MODIS and VIIRS FPAR time series (2018-2023),
we suggest to start using VIIRS from the start of its availability (2018) to ensure the maximum coherence of FPAR between
the latest years and NRT data.

## 9    Data availability

NRT data are available within 24 hours from the end of each dekad, typically at 12:00 UTC+2 of the day 1, 11 and 21
of each month. The filtered FPAR timeseries of both MODIS and VIIRS can be accessed through the public JRC Data
Catalogue. The MODIS-FPAR filtered dataset is available at the following persistent identifier https://data.jrc.ec.europa.eu/

dataset/1aac79d8-0d68-4f1c-a40f-b6e362264e50 (Seguini et al., 2025) or can be directly downloaded from the following server https://agricultural-production-hotspots.ec.europa.eu/data/MO6_FPAR/. The server data structure is divided according to the data source with one folder dedicated to the MODIS data and another to the intercalibrated VIIRS data. Subfolders contain the associated consolidation stage products.

The structure of the subfolders follows the year and consolidation stage order. Each geotiff has a naming convention like SSYYDDVVVCX, where SS describes the sensor (MT for MODIS and IT for intercalibrated VIIRS), YY the year, DD the dekad of reference, VVV the name of the product and X the stage. As example it1819FPRC1 indicates the consolidation stage C1 from the intercalibrated VIIRS-FPAR filtered data for year 2018 and dekad 19. Associated to each Geotiff raster a text files contains all the reference metadata. An example of the metadata file is reported in Table E1.

To facilitate product exploration, rasters and statistics at regional level (including temporal trajectories aggregated over cropland and rangeland areas) can be visualized in the online Warning Explorer of the ASAP system (https://agricultural-production-hotspots.ec.europa.eu/wexplorer).

## 10   Conclusions

We released a dataset of two timeseries, from MODIS and VIIRS, of global FPAR data at 500 m resolution, updated every 10 days since 2000. This dataset is optimized for agricultural applications, including NRT monitoring of biomass productivity of cropland and rangeland, and crop yield forecasting. We tuned the filtering parameters using cropland pixels from Europe and Africa and applied specific algorithms (e.g., constraint mechanisms and gap-filling) for periods with only low-quality observations. As demonstrated in the ablation study (Section 4.2.2), applying constraints to early FPAR estimations (C0 to C4) with few or no HQ observations leads to better alignment with the consolidated CF value, compared to unconstrained estimations. To avoid unrealistically extrapolated FPAR values in absence of HQ information, our estimations are set to be conservative, relying on historic information (i.e., gap-filling and constraint approaches). This approach guarantees that the NRT value of FPAR is always estimated, thus allowing subsequent analysis that otherwise could be potentially hampered by missing data. Our dataset thus meets the operational needs of early warning systems (EWS) and crop yield forecasting systems (CYFS), serving accurate FPAR estimation instead of missing information. In addition to NRT requirements, EWS and CYFS require long timeseries to capture interannual variability in crop growth. With the MODIS sensor mission nearing its end, and ongoing acquisition issues since 2022, we have generated a VIIRS-FPAR timeseries, corrected over the MODIS-FPAR data, to extend the record into the coming years. Although continuity between MODIS Collection 6.1 and VIIRS Collection 2 FPAR timeseries was planned and confirmed, we still observed some discrepancies. A simple correction procedure (MD) significantly reduced these discrepancies across various consolidation stages and regions, though small differences remain. With the recent release of the VIIRS-FPAR Collection 2 data from 2012 onward, more advanced intercalibration methods are currently being tested, benefiting from greater overlap between MODIS-FPAR and VIIRS-FPAR data. The JPSS program ensures the sustainability of this processing pipeline, with VIIRS-NOAA-21 now operational and plans for two more satellites (JPSS-3 and JPSS-4) to ensure continuity through the early 2030s. NRT gap-filled data provision and long term data records can

find applications beyond EWS and CYFS, e.g. in EO-based Index Insurance programs (De Leeuw et al., 2014) where seasonal FPAR anomalies are used as the index to determine payouts. We recommend data users to utilize the MODIS-FPAR filtered timeseries from 21-08-2000 to 31-12-2017 and from 01-01-2018 onward the intercalibrated VIIRS-FPAR filtered timeseries. We plan to keep the operational products updated for the changing landscape of data availability in response to the foreseen availability of other VIIRS sensors in the near future. To serve the community, data are, and will remain, freely available.

## Appendix A:  QA Layer information used

**Table A1.** The MODIS FPAR QA layer information used in the filtering process (Myneni, 2020)

| Layer name | Bit N° | Parameter name | Bit | Description | Filtering usage |
|---|---|---|---|---|---|
| FparLai_QC | 1 | Sensor | 0 | Terra | Additional information |
| | | | 1 | Aqua | |
| | 2 | Dead Detector | 0 | Detectors apparently fine for up to 50% of channels 1, 2 | Exclude invalid observations |
| | | | 1 | Dead detectors caused >50% adjacent detector retrieval | |
| | 3 - 4 | Cloud state | 00 | Significant clouds NOT present (clear) | Downweight unreliable observations |
| | | | 01 | Significant clouds WERE present | |
| | | | 10 | Mixed cloud present in pixel | |
| | | | 11 | Cloud state not defined, assumed clear | |
| | 5 - 7 | SCF_QC | 000 | Main (RT) method used, no saturation | Downweight unreliable observations |
| | | | 001 | Main (RT) method used with saturation | |
| | | | 010 | Empirical algorithm due to bad geometry | |
| | | | 011 | Empirical algorithm due to other problems | |
| | | | 100 | Pixel value not produced at all | |
| FparExtra_QC | 2 | Snow or ice | 0 | No snow nor ice detected | Downweight unreliable observations |
| | | | 1 | Snow or ice detected | |
| | 3 | Aerosol | 0 | No or low atmospheric aerosol detected | Downweight unreliable observations |
| | | | 1 | Average or high aerosol levels detected | |
| | 4 | Cirrus | 0 | No cirrus detected | Downweight unreliable observations |
| | | | 1 | Cirrus detected | |
| | 5 | Internal cloud mask | 0 | No clouds | Downweight unreliable observations |
| | | | 1 | Clouds detected | |
| | 6 | Cloud shadow | 0 | No cloud shadow detected | Downweight unreliable observations |
| | | | 1 | Cloud shadow detected | |

**Table A2.** The VIIRS FPAR QA layer information used in the filtering process (Park et al., 2018b)

| Layer name | Bit N° | Parameter name | Bit | Description | Filtering usage |
|---|---|---|---|---|---|
| FparLai_QC | 0 - 2 | SCF_QC | 000 | Main (RT) method used, no saturation | Downweight unreliable observations |
| | | | 001 | Main (RT) method used with saturation | |
| | | | 010 | Empirical algorithm due to bad geometry | |
| | | | 011 | Empirical algorithm due to other problems | |
| | | | 100 | Pixel value not produced at all | |
| | 2 | Dead Detector | 0 | Both red and NIR detectors are fine | Exclude invalid observations |
| | | | 1 | At least one band has dead detector | |
| FparExtra_QC | 0 - 1 | Cloud detection and confidence | 00 | Confident clear | Downweight unreliable observations |
| | | | 01 | Probably clear | |
| | | | 01 | Probably cloudy | |
| | | | 01 | Confident cloudy | |
| | 2 | Cloud shadow | 0 | No cloud shadow | Downweight unreliable observations |
| | | | 1 | Shadow | |
| | 3 | Thin cirrus | 0 | No cirrus detected | Downweight unreliable observations |
| | | | 1 | Cirrus was detected | |
| | 4 - 5 | Aerosol quantity | 00 | Climatology | Downweight unreliable observations |
| | | | 01 | Low | |
| | | | 10 | Average | |
| | | | 11 | High | |
| | 6 | Snow/Ice | 0 | No | Downweight unreliable observations |
| | | | 1 | Yes | |

**Appendix B: Sampling**

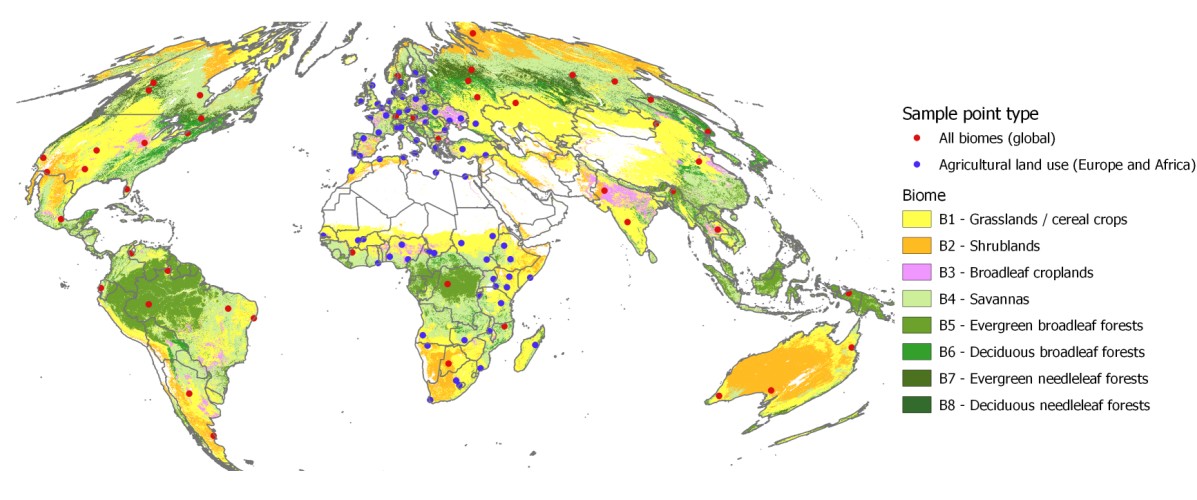

**Figure B1.** Sample points used to find suitable settings for weighting and implementing the Whittaker filter at the global level (red to represent all biomes) and in Europe and Africa (blue to focus on agricultural land use); biome data (Friedl and Sulla-Menashe, 2019)

**Table B1.** Sample points distribution according to their geographical position and biome

| Biome | Europe | Africa | Global | Total |
|---|---|---|---|---|
| B1 Grasslands / cereal crops | 25 | 21 | 12 | 58 |
| B2 Shrublands | - | - | 5 | 5 |
| B3 Broadleaf croplands | 9 | 12 | 5 | 26 |
| B4 Savanna | 1 | 2 | 9 | 12 |
| B5 Evergreen Broadleaf Forests (EBF) | - | - | 4 | 4 |
| B6 Deciduous Broadleaf Forests (DBF) | - | - | 4 | 4 |
| B7 Evergreen Needleleaf Forests (DNF) | - | - | 6 | 6 |
| B8 Deciduous Needleleaf Forests (DNF) | - | - | 2 | 2 |
| Total | 35 | 35 | 47 | 117 |

## Appendix C: QA Layers produced

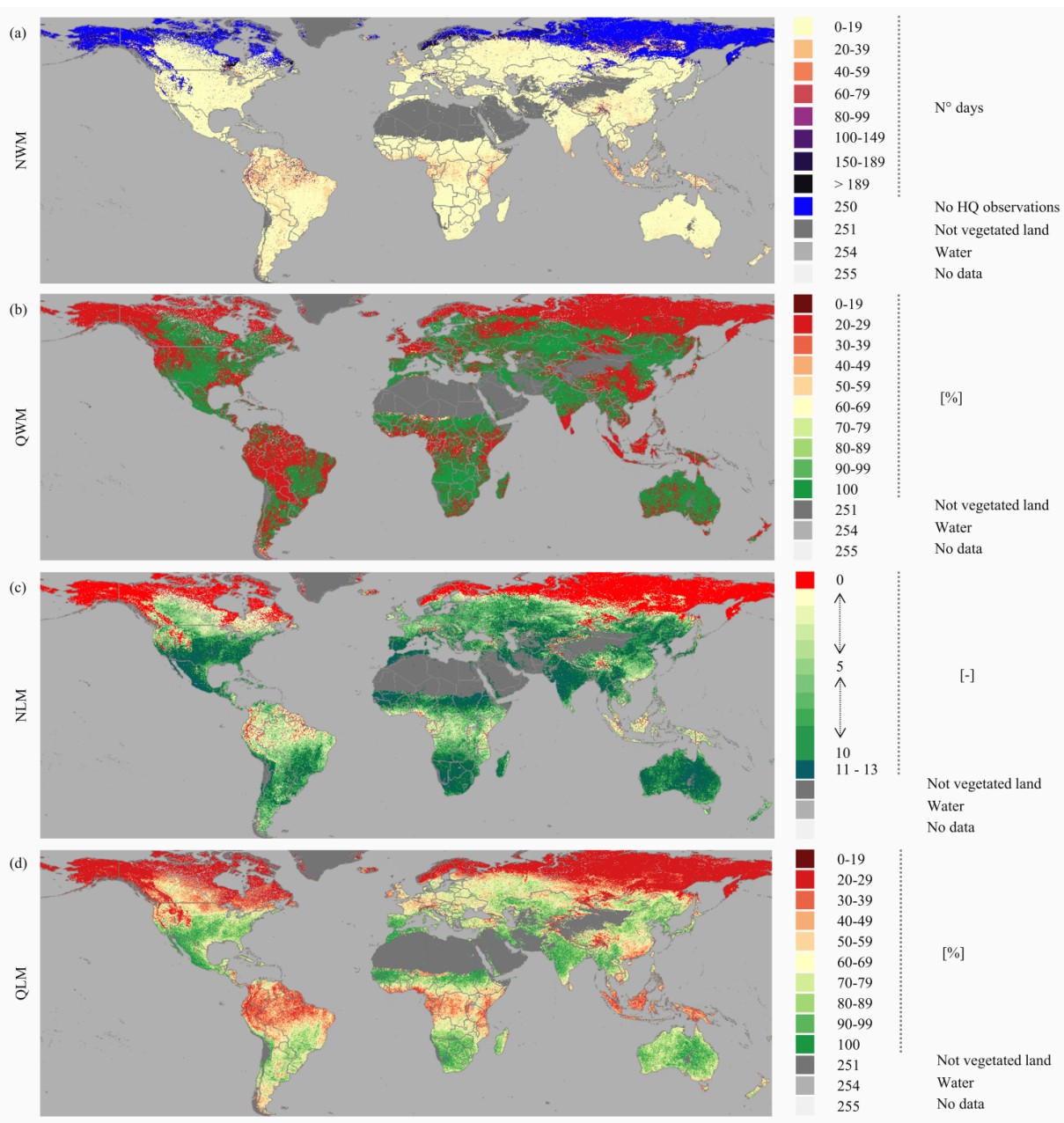

**Figure C1.** Examples of QA Layers produced during the filtering of MODIS FPAR as described in 2 for the period 1-10 May (dekad 13) 2023

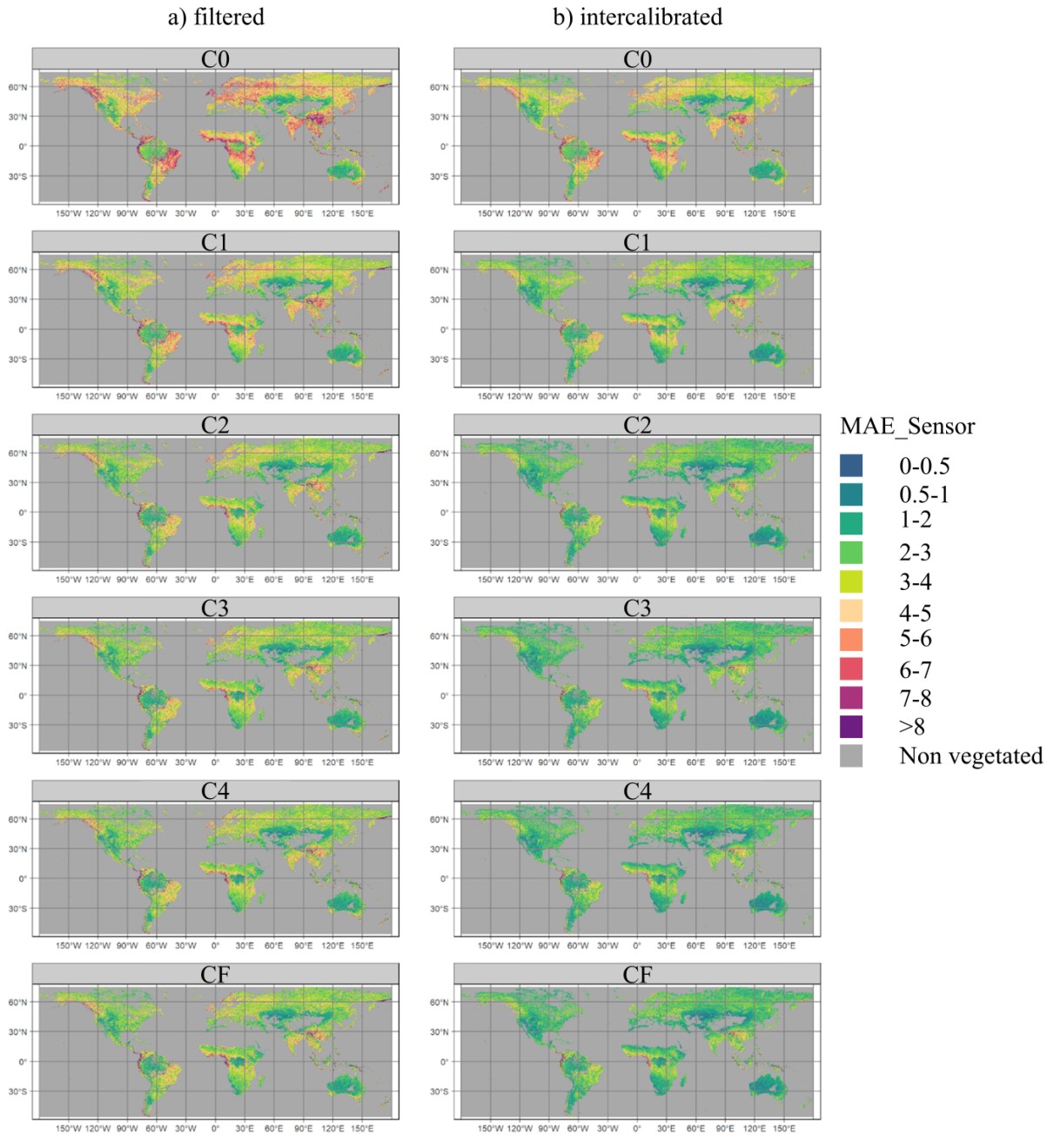

**Figure C2.** Geographical distribution of MAD according to the consolidation stage FPAR for vegetated pixels during the growing season (201819-202318). MAD is computed in reference to the MODIS-FPAR filtered timeseries using VIIRS-FPAR filtered timeseries (left) or intercalibrated VIIRS-FPAR filtered timeseries (right).

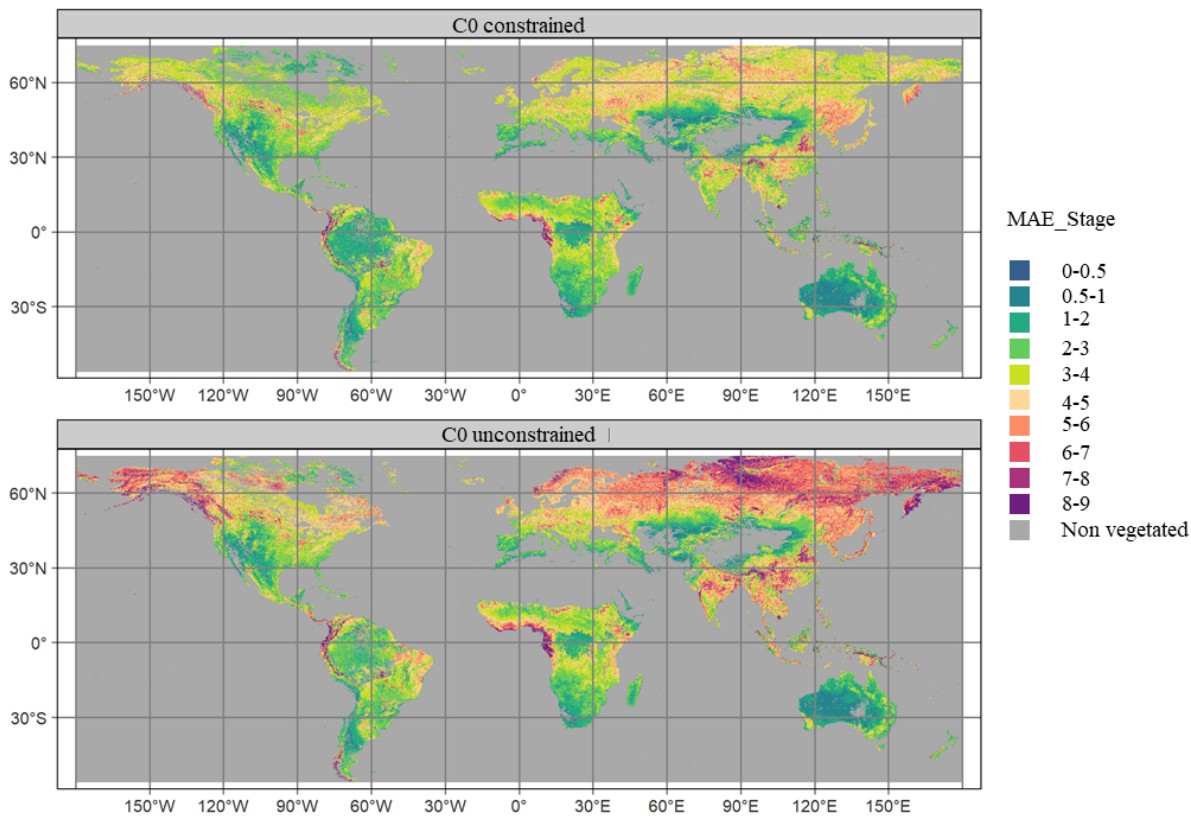

**Figure D1.** The upper panel shows the MAE_Stage computed for C0 stage from the MODIS-FPAR filtering, using the constraint mechanism. The lower panel displays the same metric computed without using constraints. FPAR values are expressed in %.

## Appendix E: Metadata

**Table E1.** Filed name and description of the metadata of a FPAR global raster

| Filed name | Description |
| --- | --- |
| Driver | Type of driver used to produce the file |
| File | Original location of the file |
| Size | Width and height in n°of pixel |
| Coordinate System is: | Type of coordinate systems expressed in EPSG code |
| Origin | Coordinates of the raster origins |
| Pixel size | Width and height of each pixel expressed in the reference unit |
| AREA_OR_POINT | Type of information |
| consolidation_period | Period of consolidation according to the filtering nomenclature (from C0 to CF) |
| file_creation | Date of creation of the file |
| flags | Flags used |
| highest_actual_value | The highest value in the file |
| highest_possible_value | The upper maximum value accepted |
| lineage | File name |
| lowest_actual_value | The minimum value in the file |
| lowest_possible_value | The minimum maximum value accepted |
| program | The version of the program used |

Below, in italics an example of metadata associated to the intercalibrated VIIRS-FPAR file for the consolidation stage C1 for the period 1-10 May (dekad 13) 2023.

---

*Driver: GTiff/GeoTIFF*

*Files: /vitodata/Mars/MEP/MVIIRS/V070/GLO/ACT/IMG/2018/it2313FPRC1.tif*

*Size is 80640, 29346*

*Coordinate System is:*

*GEOGCS["WGS 84",*

*DATUM["WGS_1984",*

*SPHEROID["WGS 84",6378137,298.257223563,*

*AUTHORITY["EPSG","7030"]],*

*AUTHORITY["EPSG","6326"]],*

*PRIMEM["Greenwich",0],*

*UNIT["degree",0.0174532925199433],*

*AUTHORITY["EPSG","4326"]]*

*Origin = (-180.004464285714988,75.004464285715002)*

*Pixel Size = (0.004464285715000,-0.004464285715000)*

*Metadata:*

*AREA_OR_POINT=Area*

*consolidation_period=C1*

*creator=VITO*

*date=20230501*

*days=10*

*description=MODIS/VIIRS, FPAR, Smoothed 500m, Product-version=V070*

*file_creation=2024-03-25T10:41:43*

*flags=251=other land, 254=water, 255=not processed*

*highest_actual_value=100*

*highest_possible_value=100*

*lineage=it2313FPRC1.tif*

*lowest_actual_value=0*

*lowest_possible_value=0*

*program=0.1.1*

---

500 *Author contributions.* Writing: AK, LS. Conceptualization: MM, AK, LS. Review & editing: AV, GM, FR, CA

*Competing interests.* The authors declare that they have no conflict of interest

*Acknowledgements.* This research has been supported by the MarsOP6 activity of the Agri4cast project (Joint Research Center of the European Commission). The computational results presented have been partially achieved using the Vienna Scientific Cluster (VSC)

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
