# Peer review of "Global near real-time 500 m 10-day FPAR dataset from MODIS and VIIRS for operational agricultural monitoring and crop yield forecasting"

_Earth System Science Data, 2025_

## Author Comment (AC3)

**RC3**: *This paper presents a comprehensive methodology for generating a global near real-time (NRT) FPAR dataset at 500m resolution with 10-day temporal resolution, combining MODIS and VIIRS satellite data for agricultural monitoring and crop yield forecasting applications. The authors employ a Whittaker smoother-based filtering approach with quality-based weighting and constraint mechanisms to handle sparse observations due to cloud cover. The dataset provides multiple consolidation stages (C0 to CF) representing increasing data quality over time, and includes intercalibrated VIIRS-FPAR data to ensure continuity beyond the MODIS mission lifetime. The work addresses an important operational need for consistent, gap-filled vegetation monitoring data for early warning systems and crop yield forecasting. Nevertheless, there are still improvements for this manuscript.*

**Authors:**

We thank Kai Yan for reading our paper and providing relevant suggestions.

**RC3**: *1. This work presents a valuable operational implementation of established techniques, specifically for agricultural monitoring. I'm curious about the specific adaptations made to the Whittaker smoother approach compared to previous implementations. Could the authors provide more detail on what aspects of their constraint mechanisms or gap-filling procedures are novel or represent improvements over existing methods? This would help readers better understand the technical contributions.*

**Authors:**

We appreciate the opportunity to highlight the contributions we made in this work as compared to previous implementations of the Whittaker smoother. Specifically, we:

- provide adaptations for near real time requirements (i.e., using only data prior to the filtering date to smooth the timeseries);

- use the fit to the upper envelope to reduce noise from cloud contamination, thereby combining the Whittaker filter with the iterative upper envelope fit proposed by Chen et al., 2004.

If the reviewer refers specifically to the changes compared our previous implementation of Whittaker, as from Klisch and Atzberger (2016), we would like to point out that in this study we:

- revised the tuning parameters of the previous implementation to improve the fitting of the FPAR timeseries (as described in Sections 4.1.2 and 4.1.3);
- introduced the fit to the upper envelope of the FPAR temporal trajectories using the iterative approach of Chen et al. (2004) to minimize the noise introduced by the undetected cloudy observations (as described in Section 4.1.2);
- revised the weights used by the smoother based on MODIS quality indicators (as described in Section 4.1.1);
- introduced weights for VIIRS data to be consistent with the MODIS weighting approach (as described in Section 4.1.1);
- introduced the constraint activation based on LTA data in case of sparse observations (as described in Section 4.1.4);
- perfected the system of output quality layers (as described in Section 4.1.5).

According to reviewer's suggestion, we will change Section 4.1 to highlight the improvements as compared to standard Whittaker smoothing.

*"Our approach builds on the previous method for developing a NRT operational MODIS NDVI product (Klisch and Atzberger, 2016; Meroni et al., 2019) based on the Whittaker smoother (WS Atzberger and Eilers, 2011a, b). In our implementation of the WS, the FPAR observations are weighted according to the QA products (Section 4.1.1), while all the available observations are used. Compared to the previous implementations we revised the weights assigned to MODIS data and introduced new weights for VIIRS data. The WS achieves a balance between the fidelity to the original data and the roughness of the smoothed curve (i.e., the second-order differences) by tuning its smoother parameter λ: a larger λ results in smoother results that align less with original original data. In Section 4.1.2 we describe the tuning of the parameter λ and the smoothing of the whole timeseries (i.e., off-line smoothing) for the computation of the long-term statistics. Compared to the previous implementation we revised the λ parameters and introduced an iterative upper envelope fit approach to minimize noise from undetected cloudy observations (Chen et al., 2004). The operational NRT filtering is described in Section 4.1.3 while its adaptation in presence of sparse observations is presented in Section 4.1.4. Finally, in 4.1.5 we describe the quality products associated with the NRT filtering. Compared to the previous implementation, this study modified the adaptation strategy and perfected the system of the quality layers.*

**RC3**: *2. The quality layers provide valuable information about processing conditions, but I'm curious about how measurement uncertainties from the original MODIS and VIIRS products affect the final filtered results. This information would be particularly valuable for users needing to understand confidence levels in their applications.*

**Authors:**

As the reviewer highlights, we used the quality assessment (QA) products to weight our data in agreement with the MODIS user manual, which states that QA should be used to address pixel quality (Myneni, 2020). We initially evaluated the use of the MODIS retrieval uncertainty (i.e., the layer with the standard deviation (SD) values associated with each data point) and we finally considered it unsuitable for our weighting approach for two reasons. First, no uncertainty is provided when the backup algorithm is activated. As our approach uses also backup algorithm outputs (Section 4.1.1), the use of uncertainty in the weighting scheme would not be applicable. Second, we empirically noticed that uncertainties did not provide more information than the QA and at times provided contrasting information. Fig. R1 provides an example of our initial exploratory analysis performed over hundreds of randomly extracted samples of FPAR temporal profiles, for which the quality flags provide more informative value for our weighting scheme than the uncertainty estimates. In the figure, observations labelled *A* exhibit very low uncertainty but were flagged as cloudy. Observations labelled *B* have small uncertainty, but their FPAR values are flagged as saturated. In cases *A* and *B,* the quality flags offer a more valuable information for our weighting scheme than the uncertainty. Observations labelled *C* present small uncertainty yet unrealistic low FPAR values, that is, much lower than the previous and the following observations, very likely due to cloud presence. In this case uncertainty does not offer a valuable score to properly assess the observation quality. Finally, observations labelled *D* present, in a restricted time span over which we can assume little variation in FPAR, plausible and similar FPAR values, yet exhibit very different uncertainties, suggesting that uncertainty cannot be used for quality screening. The above examples describe a recurrent behaviour we observed in our exploratory analysis.

To conclude, we considered the use of uncertainty, but we deemed them unsuitable for our approach because: 1) they are not available for the back-up algorithm, and 2) their informative content seems to overlap, and at times be in contrast, with the quality assessment layers.

[Figure]

*Figure R1. FPAR MODIS data (MOD15A2H.061 and MCD15A3H.061) and associated quality information for an arable land pixel in southern Finland (point ID116 of Latitude 60.952° N and Longitude 22.982° E), for the period 2017 - 2022. Top panel: the symbols and colours represent the information available in the FparLai_QC layer of MODIS data. Symbols represent the snow and clouds flags. Colours represent the algorithm path information retrieved (SCF_QC). RT indicates that the main algorithm (radiative transfer model) was used, either saturated or retrieved under the best condition. No RT means that the back-up algorithm was used. Bottom panel: symbols represent the presence of aerosols (available in the FparExtra_QC layer) while colours represent the uncertainties values associated with the main algorithm (grey colour for points with no uncertainty computed) and available in the FparStdDev_500m layer.*

**RC3***: The evaluation metrics are appropriate. I wonder if seasonal or regional performance variations might provide additional insights into the method's behaviour under different conditions.*

**Authors:**

We appreciate the reviewer's acknowledgment of the appropriateness of the metrics we used. Indeed, we performed both regional and temporal analysis of performances, but we decided not to include it in the manuscript for conciseness.

To explore spatio-temporal variations, we produced the Hovmöller diagrams of MAE_Stage_Cx (Section 4.2.1) for MODIS-FPAR. In Fig. R2 we report the MAE_Stage_C0, as an example showing the largest discrepancies between consolidation stages, i.e. C0 and CF. Fig. R2 shows that the largest errors are typically occurring in croplands at the season peak, in June in the northern hemisphere (dekad 15 to 18) and between February and March in southern hemisphere (dekad 33 to 9). This behaviour suggests that our approach is conservative when the FPAR curve experiences rapid slope changes (for example, near the seasonal peak when the slope shifts from positive to negative, as shown in by C0 timeseries in Fig. 4 of the original paper). In such cases, C0 tends to preserve the previous FPAR trajectory, given the absence of information after the most recent filtering point.

In vegetation (i.e., all processed pixels, see Section 3) or rangeland pixel seasonality is less pronounced and a conservative FPAR estimation results in lower error.

[Figure]

*Figure R2: Hovmöller diagrams for MAE_Stage_C0 computed over MODIS-FPAR data considering the period 2003 - 2022. Diagrams are produced extracting MAE_Stage_C0 over croplands (left panel), rangelands (central panel) and all vegetated areas (right panel).*

We also analysed the spatio-temporal effects of the intercalibration. For this purpose, we produced Hovmöller diagrams for C0 before and after the intercalibration (Fig R3 below). Largest discrepancies between FPAR from the MODIS and VIIRS timeseries (Fig.R3, top panel) occurs at tropical latitudes during wet season (dekad 21-27 for latitudes around 20°, dekad 30-9 for latitudes <-20°), or at higher latitudes during the winter months (dekad 33-6 for latitudes >40°, and dekad 15-24 for latitudes <-40°). These periods of the year are associated with persistent cloud coverage (and low sun angles at high latitudes) resulting in larger data gaps. We empirically observe that under such conditions, the differences between MODIS and VIIRS FPAR estimation increases. The MD correction we applied efficiently reduces such difference as the reduction in MAE_Sensor (Fig.3, bottom panel) shows.

[Figure]

*Figure R3. Hovmöller diagrams of MAE_Sensor_C0 error before (top panel) and after (bottom panel) the MD correction for the period 2020 dekad 19 - 2023 dekad 18. In this case we focus on the growing period (i.e. off-season period is masked out).*

**RC3***: **Minor Comments***

*The paper is generally well-written with clear explanations of the methodology. There are a few minor issues (e.g., "reangeland" should be "rangeland" in several places) that could be addressed during revision.*

*More justification for the λ=3000 parameter selection would be helpful*

*I note that "hindacsting" in line 215 should be corrected to "hindcasting" and "poral resolution" in line 90 to "temporal resolution". Please check for similar cases.*

**Authors:**

We thank the reviewer for spotting these typos. We checked for other occurrences and corrected them in the revised version of the manuscript.

As mentioned in the text, the selection of the λ value was based on our experience in previous studies (Atzberger et al., 2015; Klisch and Atzberger, 2016) and a further verification procedure that involved visual inspection of observations and fitted curves at hundreds of sample points.

Figure R4 show an example of tested λ values, but also different numbers of iterations, including an optimized number of iterations with a fitting-effect index (Chen et al., 2004).

[Figure]

*Figure R4. Example of test with different λ (a) and iteration values (b) on MODIS data. Label "chen 3" stands for the application of the fitting-effect index as from Chen et al. (2004) and the resulting number of iterations (3 in this case).*

**References**

Atzberger, C., Eilers, P.H.C., 2011a. A time series for monitoring vegetation activity and phenology at 10-daily time steps covering large parts of South America. International Journal of Digital Earth 4, 365–386. https://doi.org/10.1080/17538947.2010.505664

Atzberger, C., Eilers, P.H.C., 2011b. Evaluating the effectiveness of smoothing algorithms in the absence of ground reference measurements. International Journal of Remote Sensing 32, 3689–3709. https://doi.org/10.1080/01431161003762405

Atzberger, C., Vuolo, F., Klisch, A., Rembold, F., Meroni, M., Mello, M.P., Formaggio, A., 2015. Agriculture, in: Land Resources Monitoring, Modeling, and Mapping with Remote Sensing. CRC Press.

Klisch, A. and Atzberger, C.: Operational Drought Monitoring in Kenya Using MODIS NDVI Time Series, Remote Sensing, 8, 267,550, https://doi.org/10.3390/rs8040267, number: 4 Publisher: Multidisciplinary Digital Publishing Institute, 2016.

Chen, J., Jönsson, P., Tamura, M., Gu, Z., Matsushita, B., and Eklundh, L.: A simple method for reconstructing a high-quality NDVI time-series data set based on the Savitzky–Golay filter, Remote Sensing of Environment, 91, 332–344, https://doi.org/10.1016/j.rse.2004.03.014, 2004.

Myneni, R.: MODIS Collection 6.1 (C6.1) LAI/FPAR Product User's Guide, https://lpdaac.usgs.gov/documents/926/MOD15_User_Guide_V61.pdf, 2020

---

## Referee Report (RR1)

**Reviewer Comments**

**1. Introduction**

- ASAP system: we suggest adding a short introduction to ASAP to highlight the need for operational, near real time products.

- Lines 95–105: When discussing the objectives, consider anticipating that the work has a global scope and start introducing the study area.

**2. Study Area**

- The study area paragraph is very short, and it mostly discusses auxiliary data. Consider expanding it with more information about the spatial extent and characteristics of the regions analyzed in the examples (Portugal and France).

- Line 145: Consider adding a figure showing the global extent

**3. Methods / Data Processing**

- Line 154: Specify the method used to resample from 250 m to 500 m. What type of aggregation was applied?

- Figure 1: Add the geometric resolution under each box referring to raster data to clearly illustrate the data processing chain from raw input to final output.

- Line 166: A method is cited with two references—clarify which one you actually follow. This issue applies to other instances, e.g., the Whittaker smoother; cite only the source used.

- Where is the paragraph that explains how the anomalies are calculated (lines 374–390)?

**4. Figures and Results**

- Figure 2: Clarify the meaning of the yellow triangles in the figure.

- Figure 4: The figure shows FPAR trends for an agricultural area in Portugal, which was not mentioned in the study area or methods. Explain why this area was selected. Consider that, being agricultural land, the field may not have crops every year, leading to low or zero FPAR (e.g., as may occur in 2025).

- Figure 4: FPAR signal: In some cases, FPAR does not drop to zero after crop harvest. Explain why this occurs.

- Figure 7: Explain why this specific focus area was selected and provide additional context in the study area section. Moreover, please also include the two initial figures with their respective values and legends, which are then used to calculate the anomaly maps.